# EFFICIENT NEURON SEGMENTATION IN ELECTRON MICROSCOPY BY AFFINITY-GUIDED QUERIES

**Hang Chen**[1]**, Chufeng Tang**[1]**, Xiao Li**[1]**, Xiaolin Hu**[1,2,3*]

1. Department of Computer Science and Technology, Institute for AI, BNRist,
Tsinghua University, Beijing 100084, China
2. Tsinghua Laboratory of Brain and Intelligence (THBI),
IDG/McGovern Institute for Brain Research, Tsinghua University, Beijing 100084, China
3. Chinese Institute for Brain Research (CIBR), Beijing 100010, China
{chenhang20,lixiao20}@mails.tsinghua.edu.cn; chufeng.t@foxmail.com
xlhu@tsinghua.edu.cn

## ABSTRACT

Accurate segmentation of neurons in electron microscopy (EM) images plays a crucial role in understanding the intricate wiring patterns of the brain. Existing automatic neuron segmentation methods rely on traditional clustering algorithms, where affinities are predicted first, and then watershed and post-processing algorithms are applied to yield segmentation results. Due to the nature of watershed algorithm, this paradigm has deficiency in both prediction quality and speed. Inspired by recent advances in natural image segmentation, we propose to use query-based methods to address the problem because they do not necessitate watershed algorithms. However, we find that directly applying existing query-based methods faces great challenges due to the large memory requirement of the 3D data and considerably different morphology of neurons. To tackle these challenges, we introduce affinity-guided queries and integrate them into a lightweight query-based framework. Specifically, we first predict affinities with a lightweight branch, which provides coarse neuron structure information. The affinities are then used to construct affinity-guided queries, facilitating segmentation with bottom-up cues. These queries, along with additional learnable queries, interact with the image features to directly predict the final segmentation results. Experiments on benchmark datasets demonstrated that our method achieved better results over state-of-the-art methods with a $2\sim3\times$ speedup in inference. Code is available at https://github.com/chenhang98/AGQ.

## 1 INTRODUCTION

Neuron segmentation in electron microscopy (EM) images has significant scientific importance for connectomics research, which enables the comprehensive reconstruction of neural connections in the nervous system and provides insights into the functioning of the brain (Winding et al., 2023; Scheffer et al., 2020; Dorkenwald et al., 2022; Peddie et al., 2022). Reconstructing large-scale structures typically involves handling vast amounts of data, ranging from terabytes to petabytes (Scheffer et al., 2020). Given the intricate three-dimensional and densely packed nature of neurons, the manual reconstruction process is extremely laborious and often requires years of effort from human experts to reconstruct an entire brain (*e.g.*, a Drosophila larval brain (Winding et al., 2023)).

In recent years, many methods (Funke et al., 2019; Lee et al., 2017; Knowles-Barley et al., 2016; Huang et al., 2022b; Chen et al., 2024) have been proposed to address the challenges of neuron segmentation by employing deep neural networks as an alternative to labor-intensive manual procedures. These methods usually follow a bottom-up segmentation paradigm, as depicted in Figure 1 (a). Initially, 3D affinities, which indicate the probabilities of neighboring voxels belonging to the same neuron, are predicted using a neural network (*e.g.*, 3D U-Net (Çiçek et al., 2016)) and then transformed into fragments via a clustering algorithm where watershed (Meyer, 1992) is the

---

*Corresponding author.

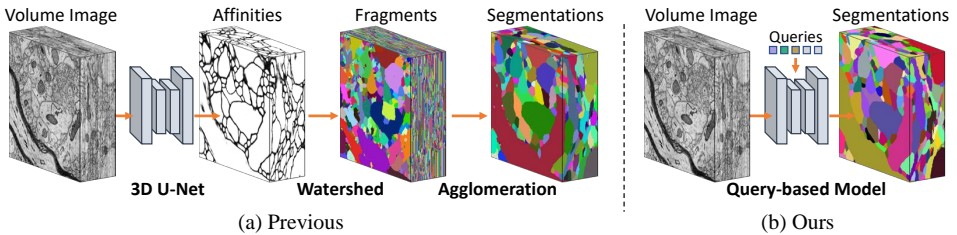

Figure 1: Comparison of the proposed model with previous neuron segmentation methods. (a) Previous methods utilize neural networks to predict affinities, and then derive the final segmentation results via watershed and agglomeration. (b) The proposed query-based model directly predicts the segmentation results, which is more concise and efficient. Best viewed in digital with zoom-in.

de-facto practice. Subsequently, these fragments are merged into final neuron segmentation results through an agglomeration algorithm (Funke et al., 2019), which merges fragments based on their affinities at interfaces. However, these methods, heavily relying on traditional clustering algorithms such as watershed, suffer from several drawbacks that limit their application: (1) The watershed algorithm usually produces fragments with unnatural boundaries such as straight lines, referred to as artifacts, due to competition between seeds, leading to inaccurate segmentation results (as shown in Section 4.4); (2) These methods may incorrectly merge different neurons and make them difficult to distinguish in the agglomeration step; (3) The watershed algorithm lacks parallelizability, making it unsuitable for GPU acceleration, resulting in inefficiency. Moreover, the large amount (*e.g.*, more than 100 thousands in a volume (Huang et al., 2022b)) of generated fragments slows down the subsequent post-processing steps such as agglomeration. Despite recent efforts to improve neuron segmentation through various techniques (Chen et al., 2023; Funke et al., 2019; Huang et al., 2022a; Liu et al., 2023b; 2022) (see Section 2), they still adhere to the traditional paradigm and, as a result, encounter the aforementioned problems. Note that there are also methods (Januszewski et al., 2018; Meirovitch et al., 2019) that work without using watershed, which we discuss in detail in Section 2.

Neuron segmentation can be considered as a form of instance segmentation which has been extensively studied in computer vision. But it has notable differences from conventional instance segmentation. Firstly, the input data involves 3D volume images rather than 2D images. Secondly, the target objects in neuron segmentation are 3D neurons rather than natural objects with distinctive appearance. Given the morphologically complex and homogeneous nature of neurons, which exhibit dense distributions and similar appearances, distinguishing different neurons presents a significant challenge. For example, region-based methods like Mask R-CNN (He et al., 2017) may encounter difficulty in addressing this problem due to the ambiguity of bounding boxes, as pointed out in Huang et al. (2022b). Recently, query-based instance segmentation methods have shown promise in natural images, offering both efficiency and accuracy advantages (Cheng et al., 2022b;a; Zhang et al., 2021). However, these methods cannot be directly applied to neuron segmentation because their specific design for 2D natural images faces problems such as unaffordable memory overhead when ported to 3D data.

In this work, we seek to explore the possibility of the query-based model in neuron segmentation, which could potentially address the limitations associated with current methods. The motivation is that the query-based model can directly predict the final segmentation results, thus avoiding the inaccuracy and inefficiency caused by watershed. However, we found that transferring the query-based methods to neuron segmentation is nontrivial. First, existing query-based models are tailored for 2D natural images and cannot be directly transferred to 3D neuron segmentation due to the unaffordable GPU memory overhead. Second, the distinct characteristics of neurons in EM, such as their homogeneous appearance and intricate morphology, which differ significantly from objects in natural images, could hinder the learning of queries. To tackle these challenges, we propose an efficient EM neuron segmentation model (Figure 1 (b)) with Affinity-Guided Query (AGQ). AGQ is constructed based on the predicted affinities from a lightweight affinity branch. It incorporates bottom-up cues, allowing the model to initiate predictions from coarse approximations rather than learning from scratch, thereby significantly reducing the learning difficulty. AGQ and learnable queries interact with the volume image features and with each other, capturing the diverse morphological characteristics of neurons and complementing the predictions of each other.

We conducted experiments on benchmark datasets AC3/AC4 (Kasthuri et al., 2015) and ZE-BRAFINCH (Kornfeld et al., 2017). The results demonstrated that our method achieved superior results in terms of both accuracy and efficiency. In comparison to state-of-the-art methods, our method achieved significantly lower errors with $200 \sim 300\%$ speedup. Note this speedup is significant considering that a volume for connectomics research could reach the $mm^3$ level (Zheng et al., 2018; Winding et al., 2023; Shapson-Coe et al., 2024) and processing $0.001mm^3$ volume requires hundreds of GPU hours plus hundreds of CPU hours(Sheridan et al., 2023).

The main contributions of the paper are summarized as follows:

- By introducing affinity-guided queries, we devise a lightweight query-based neuron segmentation method, eliminating the clustering algorithm used in traditional methods.
- Extensive experiments validated the effectiveness of our method, improving both the accuracy and efficiency of neuron segmentation over previous methods.

## 2 RELATED WORK

**Neuron Segmentation.** Previous methods mainly follow bottom-up segmentation fashion (Funke et al., 2019; Lee et al., 2017; Huang et al., 2022b). They use a neural network (*e.g.*, a 3D U-Net) to predict affinities, followed by the watershed algorithm to transform affinities into fragments and an agglomeration algorithm (Funke et al., 2019) to obtain the final results. The watershed and agglomeration algorithms require processing on the CPU, causing inefficiency. Recent advances include improving visual representation (Chen et al., 2023; Sheridan et al., 2023), devising superior loss functions (Funke et al., 2019), implementing label efficient supervision (Huang et al., 2022a; Liu et al., 2023b), employing learnable agglomeration modules (Liu et al., 2022), and improving the U-Net architecture (Luo et al., 2024; Sun et al., 2023). APViT (Sun et al., 2023) advanced U-Net by improving the ViT architecture with tailored designs such as learnable prompt base. However, these methods still follow the previous segmentation paradigm and suffer from inaccuracy and inefficiency. FFN (Januszewski et al., 2018) processes one individual neuron at a time. Despite being a new practice, FFN is two orders of magnitude slower than affinity-based methods and has similar performance (Sheridan et al., 2023). 3C (Meirovitch et al., 2019) introduced combinational encoding to process multiple neurons in parallel, but required multiple runs and could only merge those neurons, making it more of a substitute for agglomeration. Instead, the proposed method directly predict the final segmentation results via a query-based paradigm, which is concise and efficient.

**Instance Segmentation.** The mainstream methods in instance segmentation include two-stage (He et al., 2017) and one-stage ones (Tian et al., 2020; Wang et al., 2020a;b). However, these methods were not suitable for the neuron segmentation. Two-stage methods rely on boxes to distinguish instances and cannot handle dense and widely distributed neurons. One-stage methods could not guarantee recall. Recently, query-based instance segmentation algorithms (Fang et al., 2021; Cheng et al., 2022a; Zhang et al., 2021) have emerged, which directly predict objects. However, these methods cannot be directly applied to neuron segmentation because their specific design for 2D natural images faces problems such as unaffordable memory overhead when ported to 3D data. Motivated by their success in natural images, we explore a lightweight query-based model coupled with affinity-guided queries specially tailored for neuron segmentation task. It is worth noting that although object detectors like Deformable DETR (two-stage version) (Zhu et al., 2021) also involve predicted queries, we address significantly different tasks and problems, with plenty of design differences (*e.g.*, their predicted queries are implicit and do not work with learnable queries).

**3D Medical and Biological Image Segmentation.** The segmentation of organs or lesions from CT (Wang et al., 2019; Zhou et al., 2019) or MRI (Ji et al., 2022; Zeng et al., 2020) images is the primary focus of 3D medical image segmentation. These tasks mainly involve semantic-level segmentation, which differs from the instance-level segmentation discussed in this paper. Building upon the 3D U-Net (Çiçek et al., 2016), various studies have improved the segmentation quality by refining the model structure, module design (Hatamizadeh et al., 2022; 2021; Peiris et al., 2022; Shaker et al., 2024) and label efficiency (Wu et al., 2024; Zhou et al., 2023). The 3D biological image segmentation tasks in electron microscopy include: mitochondria segmentation (Pan et al., 2023; Mai et al., 2023; Lin et al., 2021), synapse detection (Lin et al., 2021) and soma segmentation (Liu

et al., 2023a). They typically follow the bottom up fashion, where a semantic-level segmentation is first predicted and then converted into a instance-level segmentation by post-processing.

# 3 METHOD

In this section, we first introduce the preliminaries involving the problem definition of EM neuron segmentation and the previous methods. We then present a brief overview of our method, including the problem formulation and the framework. Subsequently, we detail the design and function of each module, as well as the training objectives. Finally, we describe how to apply our method on the large 3D volume EM images via block assembly.

## 3.1 PRELIMINARIES

### 3.1.1 PROBLEM DEFINITION

Given a volume EM image $I \in \mathbb{R}^{D \times H \times W}$, a neuron segmentation algorithm is required to predict the neuron id of all voxels, denoted as $S \in \mathbb{N}^{D \times H \times W}$, where $D, H, W$ denote the depth, height, and width of the image, respectively. Each id in $S$ indicates the segmentation result of a specific neuron, and id $0$ indicates the background.

### 3.1.2 PREVIOUS METHODS

As shown in Figure 1(a), previous methods typically employ deep neural networks (*e.g.*, 3D U-Net) to predict affinities $A \in [0,1]^{3 \times D \times H \times W}$ first. These affinities represent the probabilities that neighboring voxels[1] belong to the same neuron (Gao et al., 2019; Ke et al., 2018; Funke et al., 2019) (see Appendix A for more explanation). Subsequently, a seeded watershed algorithm (Meyer, 1992) is applied to extract the fragments $G$ based on the predicted affinities:

$$G = \mathrm{Watershed}(A) \quad \in \mathbb{N}^{D \times H \times W}. \tag{1}$$

Finally, the fragments $G$ obtained from the seeded watershed algorithm are merged using an agglomeration algorithm based on the statistics of the affinities at their interfaces, yielding the final segmentation result:

$$S_t = \mathrm{Agglomeration}(G, A) \quad \in \mathbb{N}^{D \times H \times W}. \tag{2}$$

The agglomeration usually only merges fragments and cannot dissolve incorrect mergers. Thus, the seeds of the watershed are usually densely picked, incurring significant redundancy. On the one hand, the redundancy causes unnatural boundaries such as straight lines, referred to as artifacts, due to the competition between seeds; while on the other hand, it slows down the overall inference speed.

## 3.2 QUERY-BASED NEURON SEGMENTATION FRAMEWORK

To address the above issues, we aim to devise a concise and efficient query-based neuron segmentation method. We adopt a new modeling manner and directly predict $N$ potential segmentation probability maps $P \in (0,1)^{N \times D \times H \times W}$ based on $N$ queries. During inference, we obtain the segmentation results $S_a$ by simply taking argmax on the first dimension of $P$ to obtain the segmentation results (*i.e.*, neuron id):

$$S_a = \arg\max_{i \in 1, \dots, N} (P_{i,:,:,:}) \quad \in \mathbb{N}^{D \times H \times W}. \tag{3}$$

Without the watershed algorithm, this modeling simplifies the inference process considerably and could achieve faster inference than previous methods.

Figure 2 instantiates several query-based models, consisting of 3D U-Net backbone and decoders to predict segmentation results. We create two baselines based on two mainstream decoder design (Figure 2 (a) and (b)) from state-of-the-art query-based models on natural images (Zhang et al., 2021; Cheng et al., 2021; 2022a). These two baselines have been adapted for the 3D data of neuron segmentation to make the GPU memory usage affordable. However, we found that neither baseline

---

[1]Each voxel has six neighbors, of which only three need to be predicted due to duplication.

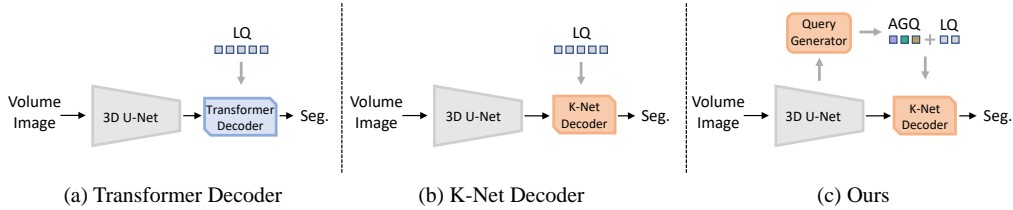

Figure 2: Comparison of different query-based models. LQ and AGQ denotes learnable queries and affinity-guided queries, respectively.

could tackle the task-specific challenges in neuron segmentation (see Section 4.5.1). The reason could be that query-based methods on natural images typically involve only learnable queries that are randomly initialized and updated by gradient descent. These queries lack inductive bias and does not discriminate well between neurons which have the same appearance and complex structure. This may underlie the difficulty encountered by these methods. In view of this, we propose to embed certain prior knowledge of affinities into the generation of queries. We chose K-Net decoder as shown in Figure 2 (c), since our experiments showed that it outperformed the Transformer decoder.

### 3.3 OVERVIEW OF THE PROPOSED METHOD

Figure 3 shows our framework, which consists of a 3D U-Net backbone network for feature extraction, a module to generate affinity-guided queries, and a 3D neuron decoder for directly predicting segmentation results. We then describe each module in detail.

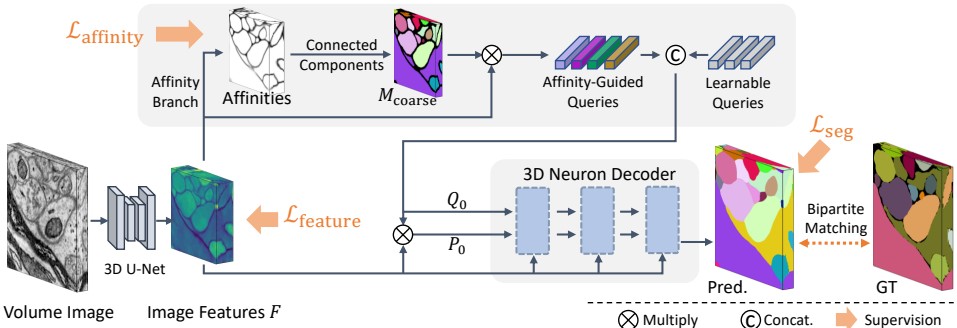

Figure 3: Overview of the proposed method. $Q_0$ and $P_0$ stand for the initial queries and segmentation probability maps, respectively. $M_{\text{coarse}}$ denotes coarse binary masks. $\mathcal{L}_{\text{affinity}}, \mathcal{L}_{\text{feature}}, \mathcal{L}_{\text{seg}}$ are losses for predicted affinities, image features and segmentation probability maps, respectively.

### 3.4 3D NEURON DECODER

Our efficient 3D neuron decoder is inspired by the query-based model for natural images (Zhang et al., 2021). The decoder comprises multiple stages with the same structure, as depicted in Figure 4. Each stage incorporates three inputs:

- The image features $F \in \mathbb{R}^{C \times D \times H \times W}$, where $C$ denotes the feature dimension [2];
- The segmentation probability maps from the previous stage $P_{i-1} \in (0, 1)^{N \times D \times H \times W}$;
- The neuron queries from the previous stage $Q_{i-1} \in \mathbb{R}^{N \times C}$, where $i$ denotes the current stage (see Section 3.5 for the definition of $P_0$ and $Q_0$).

With these inputs, we initially update the neuron queries using image features and segmentation predictions from the previous stage:

$$Q'_i = \text{DyConv}_i(Q_{i-1}, P_{i-1}F^\intercal), \tag{4}$$

---

[2]The dimension of batch size is omitted for simplicity.

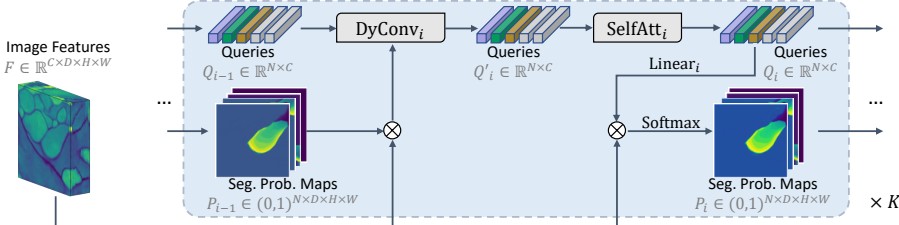

Figure 4: The detailed structure of the 3D Neuron Decoder. $K$ denotes the number of stages.

where $\text{DyConv}_i$ is a dynamic convolution layer (Fang et al., 2021), and $P_{i-1}F^{\mathsf{T}} \in \mathbb{R}^{N \times C}$. The enhanced queries can better capture the structural information of the morphologically complex and homogeneous neurons. Subsequently, these queries are processed through a self-attention layer $\text{SelfAtt}_i$ (Vaswani et al., 2017):

$$Q_i = \text{SelfAtt}_i(Q'_i), \tag{5}$$

where the queries interact with each other to better distinguish among the neurons (*e.g.*, deciding which query predicts which neuron). Finally, these queries interact with the image features to generate the segmentation probability maps for the current stage,

$$P_i = \text{Softmax}(\text{Linear}_i(Q_i)F), \tag{6}$$

where $\text{Linear}_i(Q_i)F \in \mathbb{R}^{N \times D \times H \times W}$ and softmax is applied along its first dimension. The $\text{Linear}_i$ represents a linear layer. This global and direct prediction approach can mitigate potential systematic artifacts that might arise from the watershed algorithm.

## 3.5 AFFINITY-GUIDED QUERY

To enhance the learning efficiency of the 3D neuron decoder, we propose the affinity-guided queries. These queries are designed to include coarse neuron structure information, allowing the decoder to initiate the process from it rather than from scratch. Initially, we utilize an affinity branch composed of multiple convolutional layers to predict affinities based on image features $F$,

$$A = \text{AffinityBranch}(F) \in (0,1)^{3 \times D \times H \times W}. \tag{7}$$

Subsequently, the predicted affinities are averaged along the first dimension to derive $\bar{A} \in (0,1)^{D \times H \times W}$, which is then thresholded to create a 3D binary mask. We use the mean value of $A$ across all dimensions, refered as $a$, as the threshold. This is to ensure that there is at least one foreground connected component during training. Following this, a fast connected component extraction algorithm (Wu et al., 2005) is employed to generate coarse masks:

$$M_{\text{coarse}} = \text{ConnectedComponent}(\bar{A} > a), \tag{8}$$

where $M_{\text{coarse}} \in \{0,1\}^{N_a \times D \times H \times W}$ comprises a collection of (denoted as $N_a$) boolean masks. By multiplying and averaging each of these masks with the image features, we obtain the respective affinity-guided queries $Q_{\text{affinity}} \in \mathbb{R}^{N_a \times C}$, where $C$ denotes the feature dimension. These queries encapsulate the rudimentary neuron structure details, thereby facilitating the subsequent stages and reducing their reliance on training data. This enhancement leads to improved segmentation accuracy.

Due to the possibility of incorrect merging in $M_{\text{coarse}}$, the number of affinity-guided queries could be less than the actual number of neurons, resulting in missing neurons not being predicted. Thus we also use $N_l$ learnable queries $Q_{\text{learnable}} \in \mathbb{R}^{N_l \times C}$. We concatenate them with affinity-guided queries together as the initial queries of the 3D neuron decoder (*i.e.*, $N = N_a + N_l$),

$$Q_0 = \text{Concat}(Q_{\text{affinity}}, Q_{\text{learnable}}) \in \mathbb{R}^{(N_a + N_l) \times C}. \tag{9}$$

Note that $N_a$ varies between passes through the network while $N_l$ is kept constant. The learnable queries are expected to interact with affinity-guided queries through self-attention layer (in Equation (5)) to recognize and predict the missing neurons. The initial segmentation probability maps $P_0$ are created by taking the product[3] of $Q_0$ (post a linear layer) and image features $F$, which is then passed through a softmax function, *i.e.*, $P_0 = \text{Softmax}(\text{Linear}_0(Q_0)F)$.

---

[3]The two matrices have the shapes $N \times C$ and $C \times (D \times H \times W)$, with reshaping omitted for simplicity.

## 3.6 TRAINING OBJECTIVES

During training, we employ three loss functions to supervise the predicted affinities, the image features, and the final segmentation results, respectively,

$$\mathcal{L} = \mathcal{L}_{\text{affinity}} + \mathcal{L}_{\text{feature}} + \mathcal{L}_{\text{seg}}, \tag{10}$$

where $\mathcal{L}_{\text{affinity}}$ denotes the binary cross-entropy loss, $\mathcal{L}_{\text{feature}}$ denotes the contrastive loss, and $\mathcal{L}_{\text{seg}}$ is a weighted combination of dice loss and cross-entropy loss. See Appendix B for more details.

## 3.7 BLOCK ASSEMBLY

Due to the extensive coverage of volumetric EM images across large continuous regions (*e.g.*, $100 \times 1024 \times 1024$ voxels), it is impractical to directly input the entire volume into the network. Current methods utilize a sliding-window technique, which generates affinity on individual cropped blocks (*e.g.*, $17 \times 257 \times 257$ voxels) and then assembles them together for watershed and agglomeration. The sliding stride is smaller than the block's size (typically half) to reduce border effects. In our method, the predictions of the blocks also require assembly. To achieve this, we reuse the agglomeration function, but for merging the results of distinct blocks instead of fragments. Another difference is that we do not necessitate overlap to address border effects, which is an advantage over previous methods. More details are discussed in Appendix C.

# 4 EXPERIMENT

## 4.1 DATASETS AND METRICS

**Datasets.** To validate the effectiveness of our method, we conducted experiments on the benchmark datasets AC3/AC4 (Kasthuri et al., 2015) and ZEBRAFINCH (Kornfeld et al., 2017). AC3 and AC4 are two human-labeled sub-volumes extracted from the mouse somatosensory cortex dataset (Kasthuri et al., 2015), with dimensions of $256 \times 1024 \times 1024$ and $100 \times 1024 \times 1024$, respectively. The ZEBRAFINCH dataset contains 33 volumes (approximately $150 \times 150 \times 150$), of which we used 30 volumes as a training set and 3 volumes as a test set. More details about the datasets can be found in Appendix D.

**Metrics.** We utilized the widely used Variation of Information (VOI) (Nunez-Iglesias et al., 2013) and Adapted Rand Error (Arand) (Arganda-Carreras et al., 2015) as the evaluation metrics. VOI can be further divided into two types of errors, $\text{VOI}_{\text{split}}$ and $\text{VOI}_{\text{merge}}$, which refer to the errors of segmenting a single neuron into multiple segments and merging multiple neurons into the same segment, respectively. Both VOI and Arand indicate errors, with a lower value indicating better accuracy. Further elaboration can be found in Appendix E.

## 4.2 IMPLEMENTATION DETAILS

By default, we employed a 3D neuron decoder with two stages (*i.e.*, $K = 2$ in Figure 4), comprising 100 learnable queries and a maximum of 100 affinity-guided queries. For feature extraction, we adopted a 3D U-Net with ResBlock (He et al., 2016) (*i.e.*, ResUNet (Xiao et al., 2018)). We used Adam optimizer and trained 20k iterations by default. Training was performed on $8\times$ NVIDIA 3090 GPUs and costed about 40 hours. More implementation details are provided in Appendix F.

## 4.3 QUANTITATIVE RESULTS

We first conducted experiments on the AC3/AC4 dataset and presented the results in Table 1. The results were reproduced utilizing the `pytorch_connectomics` codebase (Lin et al., 2021) (*e.g.*, ResUNet (Xiao et al., 2018) and SwinUNETR (Hatamizadeh et al., 2021)) or referenced from published papers (*e.g.*, PEA (Huang et al., 2022b) and FragViT (Luo et al., 2024)). Our method was compared with these watershed-based methods in terms of both accuracy and efficiency.

Regarding accuracy, our method generally outperformed these methods, exhibiting superior performance in metrics such as the lowest VOI and competitive Arand. This shows the strong potential of

Table 1: Results on the AC3/AC4 dataset. "-" indicates that the value is unavailable. Inference times were tested with an NVIDIA 3090 GPU and 64 Intel Xeon Gold CPUs, which represent the time (in seconds) to process the full test set. '∗' indicates results obtained from published papers (Huang et al., 2022b; Luo et al., 2024). $VOI_s$ and $VOI_m$ are short for $VOI_{split}$ and $VOI_{merge}$, respectively.

| | Metrics | | | | Inference Time | | | |
| --- | --- | --- | --- | --- | --- | --- | --- | --- |
| | $VOI_s$ | $VOI_m$ | VOI | Arand | Model | Watershed | Agg. | Total |
| ResUNet (Xiao et al., 2018) | 1.037 | 0.258 | 1.295 | 0.154 | 81.2 | 30.8 | 35.8 | 147.8 |
| SeUNet (Lin et al., 2021) | 1.031 | 0.251 | 1.282 | 0.156 | 83.1 | 30.7 | 36.0 | 149.8 |
| UNETR (Hatamizadeh et al., 2022) | 2.750 | 0.281 | 3.031 | 0.220 | 37.0 | 41.4 | 133.0 | 211.4 |
| SwinUNETR (Hatamizadeh et al., 2021) | 1.238 | **0.191** | 1.429 | 0.110 | 80.0 | 32.5 | 43.1 | 155.6 |
| LSD (Sheridan et al., 2023) | 1.448 | 0.229 | 1.677 | 0.134 | 229.9 | 29.7 | 26.5 | 286.1 |
| ML-De (De Brabandere et al., 2017) | 1.575* | 0.615* | 2.190* | 0.196* | - | - | - | - |
| SuperHuman (Lee et al., 2017) | 1.145* | 0.263* | 1.408* | 0.122* | 52.3 | 28.4 | 19.4 | 100.1 |
| MALA (Funke et al., 2019) | 1.304* | 0.242* | 1.546* | 0.120* | - | - | - | - |
| PEA (Huang et al., 2022b) | 0.852* | 0.232* | 1.084* | 0.094* | 60.2 | 37.1 | 25.4 | 122.7 |
| FragViT (Luo et al., 2024) | 0.868* | **0.191*** | 1.054* | 0.093* | >60.2 | ≃37.1 | ≃25.4 | >122.7 |
| APViT (Sun et al., 2023) | 0.767* | 0.209* | 0.976* | **0.078*** | >60.2 | ≃37.1 | ≃25.4 | >122.7 |
| AGQ (ours) | **0.677** | 0.290 | **0.967** | 0.095 | **27.6** | **N/A** | **6.1** | **33.7** |

Table 2: Results on the ZEBRAFINCH dataset. Results on three test volumes are reported. Inference times were tested with an NVIDIA 3090 GPU and 64 Intel Xeon Gold CPUs, which represent the time (in seconds) required to process all the test volumes.

| | Volume-1 | | Volume-2 | | Volume-3 | | Infer. Time |
| --- | --- | --- | --- | --- | --- | --- | --- |
| | VOI | Arand | VOI | Arand | VOI | Arand | Total |
| ResUNet (Xiao et al., 2018) | 0.272 | 0.013 | 1.967 | 0.135 | 1.003 | 0.154 | 21.1 |
| SeUNet (Lin et al., 2021) | 0.271 | 0.012 | 1.894 | 0.152 | 1.021 | 0.162 | 21.0 |
| UNETR (Hatamizadeh et al., 2022) | 3.013 | 0.403 | 3.394 | 0.529 | 2.665 | 0.360 | 17.7 |
| SwinUNETR (Hatamizadeh et al., 2021) | 0.744 | 0.206 | 2.022 | 0.154 | 0.693 | 0.035 | 34.9 |
| SuperHuman (Lee et al., 2017) | 0.268 | 0.011 | 1.867 | **0.107** | 0.721 | 0.042 | 16.6 |
| PEA (Huang et al., 2022b) | 0.208 | 0.009 | 1.715 | 0.122 | 0.534 | 0.028 | 17.4 |
| AGQ (ours) | **0.177** | **0.008** | **1.278** | 0.134 | **0.338** | **0.024** | **6.4** |

our method, as a brand new neuron segmentation method. The achievement of the lowest $VOI_{split}$ (indicating errors in predicting a neuron as multiple segments) aligns with our motivation to avoid the extensive fragments associated with the watershed algorithm.

Regarding efficiency, our method significantly surpassed previous methods, due to its concise neuron segmentation modeling and framework. Firstly, our method did not generate segmentation results from affinities, eliminating the need for overlap prediction to address border effects (as discussed in Section 3.7). As a result, the model's inference time was reduced. Secondly, with its compact modeling, our method predicted segmentation results without the need of watershed, thus entirely saved this time. Finally, despite reusing the agglomeration function (as explained in Section 3.7), our results were relatively complete with fewer regions to process, leading to significant time savings in the agglomeration step. Overall, our method achieved a speedup of about 340% over ResUNet and 260% over PEA. Regarding FragViT and APViT, the inference time was estimated, as the code was unavailable. The estimation was made due to FragViT and APViT employing the same watershed and post-processing method as PEA but with a heavier network.

On ZEBRAFINCH dataset (Table 2), our model also exhibited general better accuracy on the three test volumes, with a speedup of about 200%. For example, our method achieved significantly lower VOI on all test volume and was 230% and 170% faster than ResUNet and PEA, respectively. For deployment, our model consumed 6.5G GPU and 5.4G CPU memory during inference on these datasets, suitable for most computing devices.

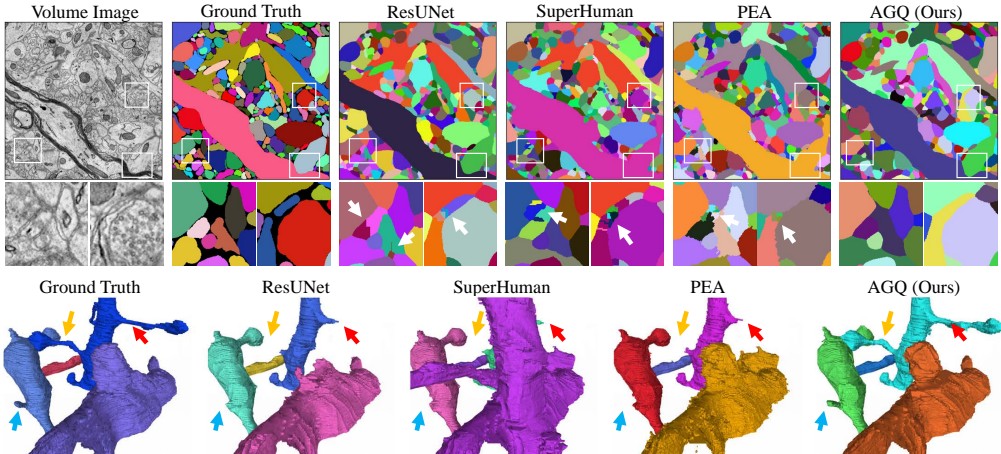

Figure 5: Visualization comparison of segmentation results obtained from different methods. The top two rows present a 2D slice alongside two zoomed-in regions, highlighting artifacts with white arrows. The third row illustrates neuron structures in 3D space, with colored arrows indicating locations that are prone to failed reconstruction. Best viewed in digital with zoom-in.

## 4.4 QUALITATIVE RESULTS

We present a qualitative comparison of segmentation results on AC3/AC4 obtained using previous methods and our method in Figure 5. The first two rows display the results on a 2D section, illustrating that our method can avoid the artifacts present in previous methods. In the second row, indicated by white arrows, the watershed-based methods generated numerous unnatural boundaries (*i.e.*, artifacts). In contrast, our method employed a learning-based approach to predict the final result directly, effectively reducing this issue.

The third row in Figure 5 showcases the 3D morphology of several predicted neurons. Highlighted by colored arrows, our method excels in reconstructing the neuronal structure. We observed that traditional methods struggle with locations characterized by thin branches or myelin sheaths, indicating the limitations of relying solely on affinities. Predicted affinities in these areas may be uncertain or distorted. Some failure cases of our method are presented and discussed in Appendix G.

To further analyze the roles of different queries in prediction, we visualized the intermediate outputs of the 3D neuron decoder in the Appendix H. As shown in Figure 10, the integration of learnable queries gradually refines the segmentation, effectively recovering missing or incorrectly merged neurons (highlighted by yellow arrows). This analysis highlights the distinct roles of the two query types: affinity-guided queries provide initial predictions, while learnable queries assist in correcting segmentation errors, particularly in resolving neuron mergers.

## 4.5 ABLATION STUDY

We conducted ablation experiments on the AC3/AC4 dataset to analyze the individual effects of each module. We report both the metrics on blocks and on the full test set (*i.e.*, before and after block assembly). Note that the models were trained for 10k iterations to save training costs, thus there is a discrepancy with the results in Table 1. More ablation experiments can be found in Appendix I.

### 4.5.1 DESIGN OF NEURON DECODER

As shown in Table 3, the K-Net style decoder achieved better results, probably due to its more explicit way of extracting instance features. Consequently, we chose the K-Net style decoder as a baseline and incorporated the proposed affinity-guided queries to achieve the best results.

Table 3: Design of neuron decoder.

| | Block | | Full | | | |
| | VOI | Arand | $\text{VOI}_{\text{split}}$ | $\text{VOI}_{\text{merge}}$ | VOI | Arand |
|---|---|---|---|---|---|---|
| Transformer Decoder | 1.245 | 0.184 | 1.695 | 1.138 | 2.833 | 0.395 |
| K-Net Decoder | 0.710 | 0.087 | 1.131 | 0.562 | 1.693 | 0.160 |
| Ours | **0.538** | **0.065** | 0.781 | 0.397 | **1.177** | **0.141** |

### 4.5.2 EFFECT OF AFFINITY-GUIDED AND LEARNABLE QUERIES

We studied the effectiveness of affinity-guided queries (AGQ) in Table 4. Completely excluding AGQ significantly deteriorated results (denoted as *LQ* in the table). Merely increasing the number of learnable queries (*i.e.*, *Double LQ*) did not lead to improved results, suggesting that enhancing segmentation quality solely by increasing query quantity is insufficient.

The effect of learnable queries (LQ) was validated by comparing the results with and without them, as shown in Table 4 (*AGQ v.s. LQ + AGQ*). The incorporation of learnable queries notably enhanced the segmentation results, particularly in mitigating merging errors (*i.e.*, $\text{VOI}_{\text{merge}}$). This finding aligned with our motivation that learnable queries could retrieve incorrectly merged neurons.

Table 4: Effect of affinity-guided queries.

| | Block | | Full | | | |
| | VOI | Arand | $\text{VOI}_{\text{split}}$ | $\text{VOI}_{\text{merge}}$ | VOI | Arand |
|---|---|---|---|---|---|---|
| LQ | 0.710 | 0.087 | 1.131 | 0.562 | 1.693 | 0.160 |
| Double LQ | 0.695 | 0.082 | 1.078 | 0.650 | 1.729 | 0.209 |
| AGQ | 0.836 | 0.102 | 0.822 | 0.881 | 2.010 | 0.229 |
| LQ+AGQ | **0.538** | **0.065** | **0.781** | **0.397** | **1.177** | **0.141** |

### 4.5.3 AFFINITY GUIDANCE

We compare different ways of leveraging affinity guidance to validate whether affinity-guided query is the optimal. Two naive baselines are concatenating affinities or coarse segmentation onto image features, which can fuse information from affinities into image features instead of queries. As shown in Table 5, AGQ yielded the best results, showing its effect in leveraging bottom-up cues.

Table 5: Results of different approaches of using bottom-up cues.

| | Block | | Full | | | |
| | VOI | Arand | $\text{VOI}_{\text{split}}$ | $\text{VOI}_{\text{merge}}$ | VOI | Arand |
|---|---|---|---|---|---|---|
| Concat affinities | 0.836 | 0.102 | 0.836 | 0.102 | 2.010 | 0.229 |
| Concat coarse segmentation | 0.869 | 0.104 | 0.975 | 0.917 | 1.893 | 0.213 |
| Affinity guided queries | **0.538** | **0.065** | 0.781 | 0.397 | **1.177** | **0.141** |

## 5 CONCLUSION AND DISCUSSION

We propose a new method named AGQ for neuron segmentation in 3D volume EM images. Diverging from previous methods that heavily rely on watershed, our method offers a more concise and efficient solution. Extensive experiments validated the effectiveness of our method, which achieved superior results and a $2\sim3\times$ speedup. We hope our method could advance the development of large-scale neuron reconstruction, thereby deepening our understanding of the brain. Future work includes exploring improved ways to handle large volumes and focused improvements for failure cases (failure cases and limitations are discussed in Appendix G).

ACKNOWLEDGMENTS

We thank Gang Zhang for helpful discussions and Lesi Wei for developing the drawing scripts. This work was supported in part by the National Key Research and Development Program of China (No. 2021ZD0200301) and the National Natural Science Foundation of China (No. U2341228).

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

## A    MORE EXPLANATIONS ABOUT AFFINITIES

The affinities indicate that whether the adjacent voxels belong to the same neuron, which have a shape of $3 \times D \times H \times W$ where $D, H, W$ is the depth, height and width of the volume image, respectively. For each voxel, these three values indicate the connection between the current voxel $(i, j, k)$ and its three neighboring voxels: $(i-1, j, k)$, $(i, j-1, k)$, and $(i, j, k-1)$, respectively. Note that due to redundancy, the voxel $(i, j, k)$ does not need to predict the connection on the other side (*e.g.*, the connection between $(i, j, k)$ and $(i+1, j, k)$ can be predicted by the voxel $(i+1, j, k)$).

The corresponding ground truth affinity value is 1 if and only if the neighboring voxels both belong to the same neuron, and 0 in all other cases (*e.g.*, belonging to different neurons or belonging to background). We present a 2D example in Figure 6 demonstrating the cases of different boundary widths between two neurons (neuron A and neuron B).

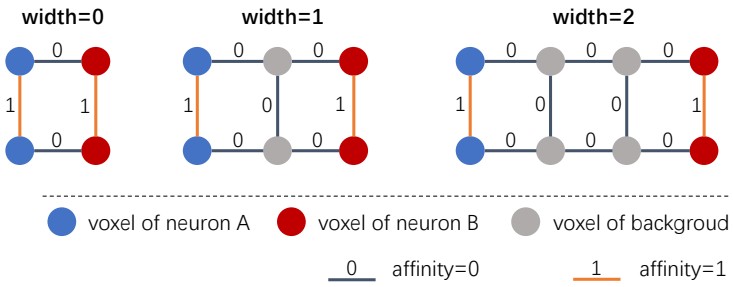

Figure 6: A 2D example demonstrating the cases of different boundary widths between two neurons (neuron A and neuron B).

## B    MORE DETAILS ON TRAINING OBJECTIVES

### B.1    SEGMENTATION LOSS $\mathcal{L}_{\text{seg}}$

To enable the end-to-end training, following DETR (Carion et al., 2020), we first identify the correspondence between predictions and the ground-truth masks via the bipartite matching and then calculate the segmentation loss based on these matched pairs. The predictions are represented as segmentation probability maps $P \in (0, 1)^{N \times D \times H \times W}$. The ground-truth masks are denoted as $M^{\text{GT}} \in \{0, 1\}^{N \times D \times H \times W}$ (padded with empty masks[4]). The optimal bipartite matching between the predictions and the ground-truth masks, symbolized as a permutation $\hat{\sigma}$ of $N$ indices, is determined using the Hungarian algorithm (Carion et al., 2020),

$$\hat{\sigma} = \arg\min_{\sigma} \sum_{i}^{N} \text{DICE}(P_{\sigma(i)}, M_i^{\text{GT}}), \tag{11}$$

where DICE denotes the dice coefficient (Milletari et al., 2016). Upon establishing the correspondence $\hat{\sigma}$, the segmentation loss is computed by combining dice loss and cross-entropy loss with weighted summation:

$$\mathcal{L}_{\text{seg}} = \lambda_{\text{DICE}} \cdot \mathcal{L}_{\text{DICE}}(P_{\hat{\sigma}}, M^{\text{GT}}) + \lambda_{\text{CE}} \cdot \mathcal{L}_{\text{CE}}(P_{\hat{\sigma}}, M^{\text{GT}}), \tag{12}$$

where the loss weights are specified as $\lambda_{\text{DICE}} = 3$ and $\lambda_{\text{CE}} = 0.3$, $P_{\hat{\sigma}}$ denotes the rearrangement of $P$ according to $\hat{\sigma}$ in the first dimension. Given that the decoder outputs multiple results from various stages (*i.e.*, $P_0, P_1, ..., P_K$ in Section 3.4), the segmentation loss is calculated individually for each output and then summed up as the total segmentation loss.

### B.2    FEATURE LOSS $\mathcal{L}_{\text{feature}}$

To enhance the differentiation among neurons in image features $F \in \mathbb{R}^{C \times D \times H \times W}$, we utilize a contrastive loss (Wang et al., 2021; Chen et al., 2022) to supervise these features. Initially, we

---

[4]Note that $N$ is usually larger than the number of neurons.

compute the summed features $T \in \mathbb{R}^{N \times C}$ for each neuron,

$$T_{i,:} = \sum_{d \leq D, h \leq H, w \leq W} M^{\mathrm{GT}}_{i,d,h,w} F_{:,d,h,w} \tag{13}$$

The features $T \in \mathbb{R}^{N \times C}$ is then normalized along the channel dimension to obtain $\hat{T} \in \mathbb{R}^{N \times C}$. Subsequently, the feature loss is computed as follows,

$$\mathcal{L}_{\mathrm{feature}} = -\lambda_{\mathrm{feature}} \sum_{d \leq D, h \leq H, w \leq W} \log \frac{\sum_i M^{\mathrm{GT}}_{i,d,h,w} \exp(\hat{T}_{i,:} \cdot F_{:,d,h,w}/\tau)}{\sum_i \exp(\hat{T}_{i,:} \cdot F_{:,d,h,w}/\tau)}, \tag{14}$$

where $\lambda_{\mathrm{feature}} = 0.1$ represents the loss weight, and $\tau = 0.3$ denotes the temperature. Notably, the calculation excludes the padded empty mask. By incorporating this loss, similar features are encouraged for voxels of the same neuron, while different features are promoted for those belonging to different neurons. This facilitates the decoder in effectively discerning between individual neurons.

### B.3 Affinity Loss $\mathcal{L}_{\mathrm{affinity}}$

The predicted affinities $A \in (0,1)^{3 \times D \times H \times W}$ were trained using binary cross-entropy loss with the ground truth $A_{\mathrm{GT}} \in \{0,1\}^{3 \times D \times H \times W}$. Note that $A_{\mathrm{GT}}$ is obtained from the segmentation ground-truth masks (*i.e.*, $M^{\mathrm{GT}}$ in Appendix B.1), which follows previous work (Lin et al., 2021; Huang et al., 2022b). The affinity loss is

$$\mathcal{L}_{\mathrm{affinity}} = \lambda_{\mathrm{affinity}} \cdot \mathcal{L}_{\mathrm{BCE}}(A, A_{\mathrm{GT}}), \tag{15}$$

where the loss weight $\lambda_{\mathrm{affinity}} = 1$. Additionally, We adopt label smoothing technique (Szegedy et al., 2016) with $\epsilon = 10^{-5}$ to prevent overfitting.

## C More Details on Block Assembly

### C.1 Full Pipeline Comparison

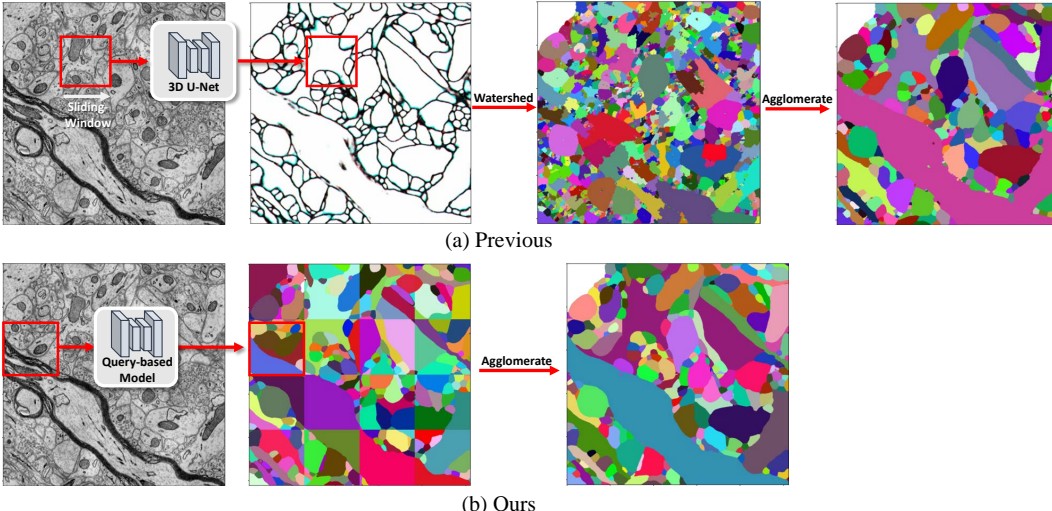

(a) Previous

(b) Ours

Figure 7: Full pipeline comparison of two neuron segmentation paradigms. (a) Previous methods utilize a sliding-window technique for inference, which generates affinities on individual cropped blocks and then assembles them together for watershed and agglomeration algorithms. (b) The proposed query-based model directly predicts the segmentation results. The results of the distinct blocks are merged by reusing the agglomerate function. Best viewed in digital with zoom-in.

In Figure 7, we present a comparison between the complete pipeline of the previous method and that of our method. Previous methods involved predicting affinities on individual blocks, which are

then assembled for watershed and agglomeration. In contrast, our method directly predicts the segmentation results on the blocks. However, assembling different blocks together poses a challenge due to the lack of natural correspondence in neuron id across blocks, unlike affinities. To address this problem, we utilize the agglomeration function to merge neurons that are separated by block boundaries. Note that we opt to reuse the agglomeration function just for implementation simplicity. Given that our task is less complex than what agglomeration typically handles (as shown in Figure 7(a)), there might be more efficient approaches to block assembly that could be explored. We show quantitatively in Table 6 the changes in metrics before and after agglomeration (*i.e.*, block assembly). Agglomeration significantly reduced split errors, due to neurons from different blocks being assembled, but also introduced merge errors to some extent.

Table 6: Results before and after agglomeration.

| Agglomeration | $VOI_{split}$ | $VOI_{merge}$ | VOI | Arand |
|---|---|---|---|---|
| Before | 3.606 | 0.181 | 3.786 | 0.874 |
| After | 0.681 | 0.267 | 0.947 | 0.089 |

## C.2 OVERLAP PREDICTION AND BORDER EFFECTS

A key difference between our method and previous methods is the significant reliance of the latter on affinities. This reliance has resulted in a common practice of predicting with overlap, such as employing a sliding window with a stride equal to half of the input size, to mitigate the impact of affinities' border effects (as depicted in Figure 8). In contrast, this strategy does not apply to our method, as we directly predict the segmentation results without relying on affinities. Table 7 provides a quantitative analysis of the effect of this strategy (*i.e.*, overlap prediction) on previous methods. On one hand, the absence of this strategy led to varying decreases in accuracy for previous methods, indicating their reliance on overlap prediction. On the other hand, even without the strategy, the previous methods were still suboptimal in efficiency due to the latency of watershed and agglomeration. This observation suggests that the efficiency improvements in our method are attributed to various factors.

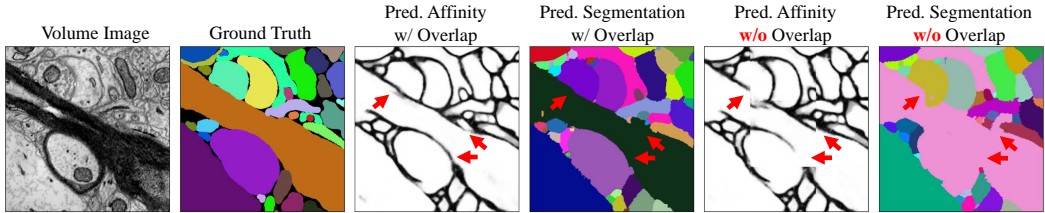

Figure 8: Illustration of the border effects. The predictions of PEA with and without overlap were compared. As highlighted by the red arrows, without overlap, the predicted affinities were notably misaligned at the block borders, leading to poor segmentation results.

Table 7: Comparison between methods with and without overlap. ✓ denotes using sliding-window with overlap, which is the default setting of previous methods. Inference times were tested with an NVIDIA 3090 GPU and 64 Intel Xeon Gold CPUs, which represent the time (in seconds) required to process the full test set (AC3).

| | | | Metrics | | | | Inference Time | | |
|---|---|---|---|---|---|---|---|---|---|
| | Overlap | $VOI_{split}$ | $VOI_{merge}$ | VOI | Arand | Model | Watershed | Agg. | Total |
| ResUNet | ✓ | 1.037 | 0.258 | 1.295 | 0.154 | 81.2 | 30.8 | 35.8 | 147.8 |
| ResUNet | | 1.075 | 0.324 | 1.399 | 0.178 | 17.6 | 38.4 | 38.2 | 94.2 |
| PEA | ✓ | 0.852 | **0.232** | 1.084 | **0.094** | 60.2 | 37.1 | 25.4 | 122.7 |
| PEA | | 1.085 | 0.583 | 1.668 | 0.218 | **11.2** | 42.7 | 27.5 | 81.4 |
| AGQ (ours) | | **0.677** | 0.290 | **0.967** | 0.095 | 27.6 | **N/A** | **6.1** | **33.7** |

## D    MORE DETAILS ON DATASETS

For AC3/AC4 dataset, following previous work (Huang et al., 2022b; Luo et al., 2024; Arganda-Carreras et al., 2015), we utilized the top 80 sections of AC4 as the training set, the subsequent 20 sections as the validation set, and the top 100 sections of AC3 as the test set for the benchmark. Each voxel in AC3/AC4 represents a physical resolution of $29 \times 6 \times 6 \ nm^3$. The ZEBRAFINCH dataset contains 33 densely labeled volumes of zebra finch brain, 29 of which have shape of $150 \times 150 \times 150$, and the other 4 of which are $128 \times 256 \times 256$. The physical resolution of each voxel is $20 \times 9 \times 9 \ nm^3$. We selected 30 of the volumes as the training set and the remaining 3 as the test set. The three test volumes are as follows, which covers both sizes:

- Volume-1: id `gt_z3734-3884_y4315-4465_x2209-2359`, shape $150 \times 150 \times 150$;

- Volume-2: id `gt_z255-383_y1407-1663_x1535-1791`, shape $128 \times 256 \times 256$;

- Volume-3: id `gt_z2868-3018_y5744-5894_x5157-5307`, shape $150 \times 150 \times 150$.

## E    MORE DETAILS ON METRICS

### E.1    VARIATION OF INFORMATION (VOI)

The VOI calculates the conditional entropy between the predicted segmentation $S_{\mathrm{Pred}}$ and the ground-truth segmentation $S_{\mathrm{GT}}$. Specifically, $H(S_{\mathrm{Pred}}|S_{\mathrm{GT}})$ reflects the amount of over-segmentation (VOI$_{\mathrm{split}}$), *i.e.*, predicting one neuron as multiple segments, while $H(S_{\mathrm{GT}}|S_{\mathrm{Pred}})$ reflects the amount of under-segmentation (VOI$_{\mathrm{merge}}$), *i.e.*, predicting multiple neurons as a single segment. The overall metric VOI is the sum of them,

$$\begin{aligned} \mathrm{VOI} &= \mathrm{VOI}_{\mathrm{split}} + \mathrm{VOI}_{\mathrm{merge}} \\ &= H(S_{\mathrm{Pred}}|S_{\mathrm{GT}}) + H(S_{\mathrm{GT}}|S_{\mathrm{Pred}}) \end{aligned} \tag{16}$$

### E.2    ADAPTED RAND ERROR (ARAND)

The Arand is calculated as:

$$\mathrm{Arand} = 1 - \frac{2 \sum_{i,j} p_{i,j}^2}{\sum_k s_k^2 + \sum_k t_k^2}, \tag{17}$$

where $p_{i,j}$ denotes the probability that a voxel is labeled as $i$ in the predicted segmentation $S_{\mathrm{Pred}}$ and $j$ in the ground-truth segmentation $S_{\mathrm{GT}}$, $s_k$ is the probability that a voxel is labeled as $k$ in $S_{\mathrm{Pred}}$, while $t_k$ is the probability that a voxel is labeled as $k$ in $S_{\mathrm{GT}}$. Voxels with id 0 in the $S_{\mathrm{GT}}$ are ignored in the calculation. In other words, Arand is one minus the harmonic average of precision and recall.

## F    MORE IMPLEMENTATION DETAILS

The learning rate followed the cosine schedule with a base learning rate of 0.0001. The total batch size is 8 (*i.e.*, one volume image block per GPU). For AC3/AC4 dataset, each block had dimensions of $17 \times 257 \times 257$. For ZEBRAFINCH dataset, each block had dimensions of $76 \times 150 \times 150$. Our code was built on the `pytorch_connectomics` (Lin et al., 2021) codebase.

In the 3D U-Net backbone, we adopted $32, 64, 96, 128, 160$ channels for its different stages. More-over, Group Normalization (Wu & He, 2018) and ELU (Clevert et al., 2015) were utilized, following the preset configurations of `pytorch_connectomics`. In the 3D neuron decoder, we used 64 channels for all hidden layers. The self-attention block consisted of 8 multi-heads, 64 channels, Layer Normalization, and GELU activation function (Vaswani et al., 2017). Additionally, we incorporated coordinate convolution (Liu et al., 2018) to enrich the image features.

Regarding the background, during training, we predict it from the image features using two convolutional layers, supervised by the same dice loss and cross-entropy loss as Equation (12). During inference, we excluded these convolutional layers to eliminate the background prediction. This practice aligns with previous methods that do not predict the background since our observations

indicated that predicting the background typically causes a decline in metrics. Moreover, during inference, neurons with a volume smaller than 40 voxels in the prediction are also excluded to filter out potential noises.

## G  LIMITATIONS AND FAILURE CASES

The representative failure cases are depicted in Figure 9. Region A showcases some redundant predictions (highlighted by the red arrows), which could be caused by multiple queries attempting to predict the same neuron simultaneously. The error observed in region B might result from a breakdown in block assembly, where parts of the same neuron in adjacent blocks were not merged correctly, leading to two disjointed segments. The error denoted by the red arrow in region C could be attributed to the ambiguity of the data, where an organelle was erroneously identified as a single neuron. To tackle these issues, potential avenues for future work could involve: (1) increasing query variability, for example, by introducing more training data or auxiliary tasks, (2) developing a dedicated block assembly algorithm for our method, and (3) collecting purposeful data for training or fine-tuning.

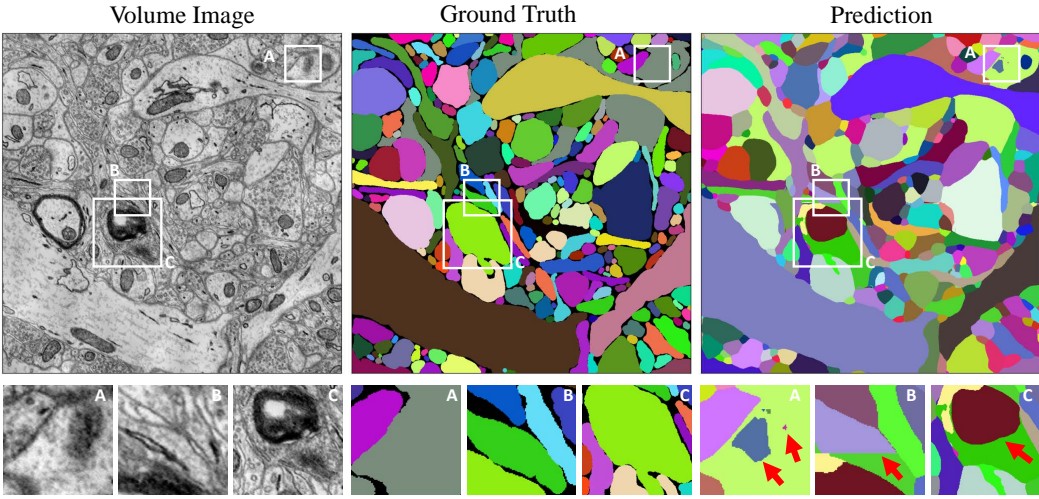

Figure 9:  Several representative failure cases. The first row shows a 2D slice. The second row presents its corresponding zoomed-in regions. The errors are highlighted with the red arrows. Best viewed in digital.

## H  ANALYSIS

To delve deeper into the roles of various queries in prediction, we visualized the intermediate outputs of the 3D neuron decoder in Figure 10. $M_{\text{coarse}}, P_0, P_1, P_2$ denote the coarse masks and results at different decoder stages, respectively (refer to Section 3.4). Each color denote a mask corresponding to a specific query. Initially, $M_{\text{coarse}}$ exhibits numerous merging errors (highlighted by yellow arrows) and only predicts a fraction of neurons. This is due to inherent errors in the predicted affinities (indicated by red arrows), such as the ambiguity in the boundaries of the myelin sheath. This also implies that relying solely on affinities as traditional methods, could raise potential issues.

Instead, our method gradually resolved these issues. As depicted by the yellow arrows in Figure 10, these missing (or incorrectly merged) neurons were progressively recovered through the integration of learnable queries. This indicates the distinct roles of the two query types: affinity-guided queries provide initial predictions, while learnable queries aid in retrieving incorrectly merged neurons.

We conducted a quantitative analysis of the segmentation results evolution in the decoder, as shown in Figure 11. Starting from coarse segmentation results, errors are gradually rectified in the 3D neuron decoder. This trend aligns with the findings in Figure 10 and underscores the importance of our multi-stage decoder design. Additionally, this highlights an advantage of our method over

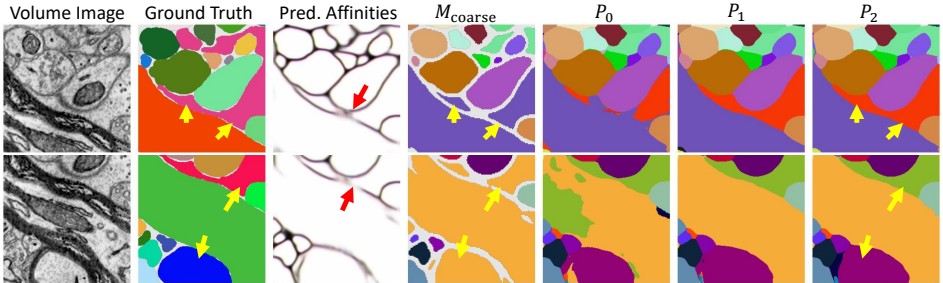

Figure 10: Visualization of the segmentation at different stages in the 3D neuron decoder. The images depict 2D sections, which inherently exist in a 3D space. Therefore, the masks in $M_{\text{coarse}}$ may interconnect in 3D. Incorrectly merged neurons in $M_{\text{coarse}}$ are highlighted by yellow arrows, whereas issues with the predicted affinities are indicated by red arrows. Best viewed in color.

previous methods: our method enables error correction through multiple stages, whereas previous methods were confined to merging results together (from watershed-generated fragments to the final results).

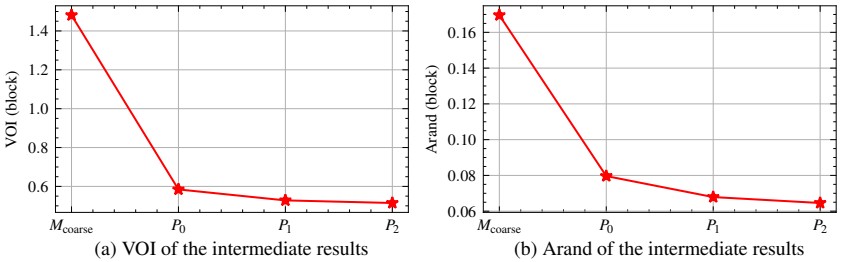

Figure 11: Benchmark of the segmentation results at different stages in the 3D Neuron Decoder. The vertical axis shows the average metrics of all blocks.

## I    MORE ABLATION STUDY

### I.1    GENERATION OF AFFINITY-GUIDED QUERIES

Table 8 compares two methods for generating the affinity-guided queries, *i.e.*, using different algorithms to derive $M_{\text{coarse}}$ from the predicted affinities. The watershed algorithm showed limited enhancement in segmentation results when utilized with the predicted affinities, possibly due to its fragmented generation of $M_{\text{coarse}}$, which offers limited support to the 3D neuron decoder. In contrast, the proposed method of thresholding the affinities and extracting the connected components led to the creation of suitable coarse masks, significantly improving the accuracy.

Table 8: Comparison of affinity-guided query generation methods.

|  | Block | | Full | | | |
|---|---|---|---|---|---|---|
|  | VOI | Arand | $\text{VOI}_{\text{split}}$ | $\text{VOI}_{\text{merge}}$ | VOI | Arand |
| Watershed | 0.706 | 0.087 | 1.122 | 0.581 | 1.703 | 0.173 |
| Connected components | **0.538** | **0.065** | 0.781 | 0.397 | **1.177** | **0.141** |

### I.2    EFFECT OF LOSS FUNCTIONS

We compared the effects of different loss functions within the proposed framework and reported the results in Table 9. The absence of feature loss notably deteriorated the segmentation results, as the image features' inability to capture neuronal specificity posed challenges for the decoder's

learning process. The exclusion of affinity loss resulted in poor model performance, given that the unsupervised affinity branch could potentially misguide the decoder. These results underscore the importance of these loss functions in achieving accurate segmentation results.

Table 9: Effect of loss functions. Feat. and Aff. denote feature loss and affinity loss, respectively.

| Feat. | Aff. | Block | | Full | | | |
|---|---|---|---|---|---|---|---|
| | | VOI | Arand | $VOI_{split}$ | $VOI_{merge}$ | VOI | Arand |
| | ✓ | 0.737 | 0.090 | 1.175 | 0.708 | 1.883 | 0.238 |
| ✓ | | 0.748 | 0.088 | 3.601 | 0.778 | 4.379 | 0.867 |
| ✓ | ✓ | **0.538** | **0.065** | 0.781 | 0.397 | **1.177** | **0.141** |

## I.3 NUMBER OF STAGES

We explored in Table 10 whether increasing the decoder's stages would enhance the segmentation accuracy. The analysis indicated that more decoder stages did not substantially improve accuracy; instead, it reached a saturation point. This observation suggests that additional stages may exceed the model's capacity beyond the constraints of the available training data.

Table 10: Number of stages. #stages denotes the number of stages.

| #Stages | Block | | Full | | | |
|---|---|---|---|---|---|---|
| | VOI | Arand | $VOI_{split}$ | $VOI_{merge}$ | VOI | Arand |
| 2 | **0.538** | **0.065** | 0.781 | 0.397 | **1.177** | 0.141 |
| 4 | 0.569 | 0.067 | 0.858 | 0.367 | 1.225 | **0.138** |

## J VISUALIZATION COMPARISON OF AXON

Figure 12 illustrates a comparison of the reconstruction results for an axon. Our method demonstrated superior reconstruction quality when compared with previous methods. As indicated by the red arrows, the previous method lost certain details, potentially attributed to the common encasement of axons in myelin sheaths and it is more difficult to predict affinities for myelin sheaths. Our method excels in reconstructing the axon, suggesting its efficacy in addressing this particular challenge.

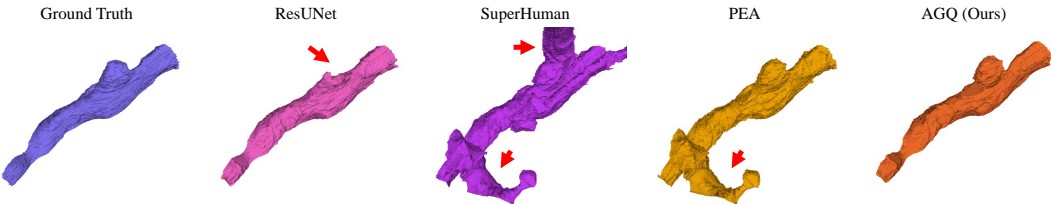

Figure 12: Visualization comparison of axon reconstructed by different methods. Red arrows indicate errors. Best viewed in digital with zoom-in.

