# OpenReview forum: "Efficient Neuron Segmentation in Electron Microscopy by Affinity-Guided Queries"
_ICLR.cc/2025/Conference — ICLR 2025 Poster_

### Official Review · Reviewer_wCU2 · 2024-10-27

**Soundness:** 3
**Presentation:** 3
**Contribution:** 2
**Rating:** 6
**Confidence:** 5

**Summary:**

The paper presents an innovative approach to neuron segmentation in electron microscopy (EM) images, a critical task in connectomics research that aims to reconstruct neural connections in the brain. The authors propose an affinity-guided query (AGQ) method, which introduces a lightweight query-based framework to address the limitations of traditional clustering algorithms.

**Strengths:**

The authors introduce AGQ to incorporate coarse neuron structure information into the query generation process, reducing the learning difficulty and improving segmentation accuracy.

**Weaknesses:**

While the work is commendable and addresses significant challenges in the field, there are areas where further development and clarification could enhance the contribution and impact of the research. Here are some specific points for consideration:

1. The paper could benefit from a more comprehensive comparison with the latest state-of-the-art (SOTA) methods, such as those referenced in [1][2]. Including these comparisons will provide a clearer picture of how the AGQ method stacks up against the most recent advancements in the field.
2. Figure 1 (b) suggests that the proposed method relies on agglomeration operations from 'waterz' post-processing. This seems to contradict the claim of directly predicting segmentation results. The authors should clarify whether the proposed method truly bypasses the need for watershed-based post-processing or if it still relies on some aspects of it.
3. The experiments presented in Table 2, while informative, may not be based on benchmarks that are widely recognized in the community. Utilizing more established benchmarks would strengthen the paper's findings and allow for better comparisons with other methods.
4. The motivation behind the proposed affinity-guided queries appears to have similarities with modules found in other works, such as [3]. The authors should elaborate on the novel aspects of their approach and how it differs from or improves upon existing techniques.
5. 'typically the neuron gap' in line168 is not unprofessional

[1]Sheridan A, Nguyen T M, Deb D, et al. Local shape descriptors for neuron segmentation[J]. Nature methods, 2023, 20(2): 295-303.
[2]Liu X, Cai M, Chen Y, et al. Cross-dimension affinity distillation for 3d em neuron segmentation[C]//2024 IEEE/CVF Conference on Computer Vision and Pattern Recognition (CVPR). IEEE Computer Society, 2024: 11104-11113.
[3]Sun R, Luo N, Pan Y, et al. Appearance Prompt Vision Transformer for Connectome Reconstruction[C]//IJCAI. 2023: 1423-1431.

**Questions:**

Comparison with State-of-the-Art Baselines

Clarification on the Use of Agglomerations

Use of Well-Established Benchmarks


Originality of Affinity-Guided Queries

---

> ### Author Response · Authors · 2024-11-23
> **Response to reviewer wCU2 (1/2)**
>
> We thank you for your valuable reviews and address your concerns as follows.
>
> **Q1**: *The paper could benefit from a more comprehensive comparison with the latest state-of-the-art (SOTA) methods, such as those referenced in [1][2]. Including these comparisons will provide a clearer picture of how the AGQ method stacks up against the most recent advancements in the field.*
>
> **A1**: Thank you for pointing out these two efforts. We have compared LSD in Table 1, which achieved sub-optimal results and was significantly slower, mainly because LSD is two-stage and consists of two models generating local shape descriptors and affinities, respectively. For CAD, after carefully investigating this work, we found that its knowledge-distillation (KD) enhanced version achieved better results, while the version without KD was less effective. However, CAD still follows the previous affinity-based paradigm and requires post-processing such as watersheds, making it even slower. For example, our overall speed is about 2.5 times the speed of CAD.
> Another fact worth clarifying is that CAD proposed a general training technique that does not conflict with our contribution. Our method could also hopefully benefit from this training technique to improve efficiency and performance even further.
>
> |         | VOI_split | VOI_merge | VOI | Arand | model (s) | watershed (s) | agglomeration (s) | total (s) |
> |---------|-----------|-----------|---------|-------|-------|-----------|------|-------|
> | CAD     | 0.601     | 0.431     | 1.032   | 0.119 | ~15.5 | ~38.5     | ~29.0| ~83.0 |
> | CAD + KD| 0.546     | 0.365     | 0.911   | 0.088 | 15.5  | 38.5      | 29.0 | 83.0  |
> | AGQ (ours)| 0.677   | 0.290     | 0.967   | 0.095 | 27.6  | N/A       | 6.1  | 33.7  |
>
> **Q2**: *Figure 1 (b) suggests that the proposed method relies on agglomeration operations from 'waterz' post-processing. This seems to contradict the claim of directly predicting segmentation results. The authors should clarify whether the proposed method truly bypasses the need for watershed-based post-processing or if it still relies on some aspects of it.*
>
> **A2**: We clarify this concern in the following two aspects:
>
> -	First, agglomeration is not a core component of the watershed algorithm. Since existing computing devices do not allow the model to process a large EM volume (e.g., 100x1024x1024) directly, for any segmentation algorithm, it is necessary to cut the volume into blocks and process them separately and then merge them. Thus, they both require an algorithm (e.g., agglomeration) to merge these results.
> -	Second, this merging can also be achieved by other algorithms, as discussed in Section C.1, better utilization of the structure prior in our method (e.g., the boundaries between blocks are all horizontal and vertical) may lead to better block assembly algorithms. We reused the agglomeration algorithm as it is compatible with existing frameworks and is relatively efficient for processing our results.
>
>
> **Q3**: *The experiments presented in Table 2, while informative, may not be based on benchmarks that are widely recognized in the community. Utilizing more established benchmarks would strengthen the paper's findings and allow for better comparisons with other methods.*
>
> **A3**: To clarify, most existing methods in the field conducted their main experiments on the AC3/AC4 dataset, which is included in the paper. For other datasets, existing methods [X1-X5] have not agreed on a consistent setting, e.g., dataset source and partition of training and test sets, etc. We appreciate your insightful suggestions. Benchmarking these methods on more datasets with consistent settings could help to better compare the advantages of these methods and ours. We will consider this as a direction for further research.
>
> [X1] Learning to Model Pixel-Embedded Affinity for Homogeneous Instance Segmentation. AAAI, 2023.
>
> [X2] Electron microscopy images as set of fragments for mitochondrial segmentation. AAAI, 2024.
>
> [X3] Self-Supervised Neuron Segmentation with Multi-Agent Reinforcement Learning. IJCAI, 2023.
>
> [X4] Cross-dimension affinity distillation for 3d EM neuron segmentation. CVPR, 2024.
>
> [X5] Large Scale Image Segmentation with Structured Loss Based Deep Learning for Connectome Reconstruction. TPAMI, 2019.

---

> ### Author Response · Authors · 2024-11-23
> **Response to reviewer wCU2 (2/2)**
>
> **Q4**: *The motivation behind the proposed affinity-guided queries appears to have similarities with modules found in other works, such as [3]. The authors should elaborate on the novel aspects of their approach and how it differs from or improves upon existing techniques.*
>
> **A4**: Thanks for pointing out this paper (referred to below as APViT). After carefully studying this work, we found that APViT still follows the previous affinity-based paradigm. APViT advances by improving the ViT architecture with tailored designs such as learnable prompt base, which is an improvement over 3D U-Net. They focus on the backbone network, while we focus on re-modeling the neuron segmentation task and designing tailored task head, which is also the main novel aspects of this paper.
>
>
> **Q5**: *'typically the neuron gap' in line168 is not unprofessional.*
>
> **A5**: Thank you for pointing this out, we have deleted this description of the paper.
>
>
> We hope you are satisfied with our clarification above. Welcome any further comments, and we'd be happy to answer them in the discussion period.

---

> > ### Comment · Reviewer_wCU2 · 2024-11-23
> >
> > Q1: I still believe that the improvement from 0.967 to 1.032 is very small. Could you explain how you replicated LSD? I am curious about whether LSD has been correctly reproduced.
> >
> > Q2: 'waterz' is a post-processing algorithm that includes waterz and agglomeration. My concern is that your method is not completely end-to-end; it’s just a simple modification of the MaskFormer series. What agglomeration algorithm are you currently using? How does it compare to the latest work in the MaskFormer series? Have you considered that this end-to-end training approach might increase the frequency of data stitching, introducing additional errors?
> >
> > Q3: In [X1-X5], Cremi is also a main experiment.
> >
> > Q4: I suggest discussing this issue in detail in the new version of the paper and providing comparisons.
> >
> > Q5: If the paper were to open-source the training and testing code, it could make a significant contribution to the community. I am particularly interested in implementing a true end-to-end segmentation algorithm.

---

> ### Author Response · Authors · 2024-11-24
> **Response to reviewer wCU2 (1/2)**
>
> Thank you for your kind and valuable feedback, which is important to us.
>
> **Q1**: *I still believe that the improvement from 0.967 to 1.032 is very small. Could you explain how you replicated LSD? I am curious about whether LSD has been correctly reproduced.*
>
> **A1**: We would like to clarify that the main contribution of this paper is to propose a new query-based paradigm for neuron segmentation task, as well as the tailored design (i.e., AGQ). Our method aims to achieve competitive or better accuracy while significantly improving efficiency and conciseness. Therefore, we emphasize more on efficiency, which is also reflected in the title. For LSD, we reproduced it with their official open-source code [X6]. We set the training set to AC4 and followed the instructions to first train the model for generating local shape descriptors (lsd) and then trained the model for predicting affinities (acrlsd), with each model trained for 200,000 iterations. For inference, we first predicted local shape descriptors on the AC3 dataset, then predicted affinities, and finally derived the final segmentation results by watershed and agglomeration.
>
> [X6] https://github.com/funkelab/lsd_nm_experiments
>
>
> **Q2**: *'waterz' is a post-processing algorithm that includes waterz and agglomeration. My concern is that your method is not completely end-to-end; it’s just a simple modification of the MaskFormer series. What agglomeration algorithm are you currently using? How does it compare to the latest work in the MaskFormer series? Have you considered that this end-to-end training approach might increase the frequency of data stitching, introducing additional errors?*
>
> **A2**: We address your concerns point by point as follows:
>
> - We clarify that we claimed the proposed method avoids the watershed algorithm and the issues it causes, instead of the specific library such as waterz.
> - We clarify that our method looks not end-to-end (Figure 8(b)) because it is impossible to feed a continuous large volume image directly into a model for training and inference (will cost unaffordable GPU memory). If the test volume fits the GPU memory, e.g. less than 17x257x257, the proposed method is completely end-to-end.  Besides, we didn't claim that our method is completely end-to-end in our paper; instead, we avoided the expression “end-to-end” in the paper.
> - We used the averaged affinities on masks' boundaries between blocks for the agglomeration, mainly to reuse existing agglomeration functions for simplicity on implementation. Other methods, such as using IoU of neighboring masks between blocks might also work.
> - We clarify that the proposed method is not a simple modification of MaskFormer series. MaskFormer and its latest work were designed for 2D tasks and would incur unaffordable GPU memory overhead if directly applied on 3D EM volume. In Section 3.2, we discussed and compared possible query-based neuron segmentation models and proposed tailored design to fit this task.
> - We argue that previous affinities-based methods also require blockwise processing (Figure 8(a)) and, since watershed algorithm is sensitive to affinities predictions, e.g., unstitched affinities lead to poor results (Figure 9), these methods even rely on overlap predictions to mitigate this issue. Compared to these works, as our method doesn't rely on watershed and is more concise, it would not introduce more such errors.
>
>
> **Q3**: *In [X1-X5], Cremi is also a main experiment.*
>
> **A3**: We clarify that [X1-X2] did not include results on CREMI dataset in their papers, while [X3-X5] differred in each other in the partition of training and test sets for CREMI dataset. [X5] trained on three volumes of 125x1250x1250, tested on three volumes of 125x1250x1250 (whose labels are private). [X3] trained on the first 60 slices of each volume, tested on the last 50 slices. [X4] trained on the first 100 slices of each volume and tested on the last 25 slices. Thus, these three works [X3-X5] did not agree on the partition of training and test sets for the CREMI dataset.
>
> [X1] Learning to Model Pixel-Embedded Affinity for Homogeneous Instance Segmentation. AAAI, 2023.
>
> [X2] Electron microscopy images as set of fragments for mitochondrial segmentation. AAAI, 2024.
>
> [X3] Self-Supervised Neuron Segmentation with Multi-Agent Reinforcement Learning. IJCAI, 2023.
>
> [X4] Cross-dimension affinity distillation for 3d EM neuron segmentation. CVPR, 2024.
>
> [X5] Large Scale Image Segmentation with Structured Loss Based Deep Learning for Connectome Reconstruction. TPAMI, 2019.
>
>
> **Q4**: *I suggest discussing this issue in detail in the new version of the paper and providing comparisons.*
>
> **A4**: Thanks to your suggestion, we have introduced this discussion and comparison in the revised paper (Section 2 and 4.3).

---

> ### Author Response · Authors · 2024-11-24
> **Response to reviewer wCU2 (2/2)**
>
> **Q5**: *If the paper were to open-source the training and testing code, it could make a significant contribution to the community. I am particularly interested in implementing a true end-to-end segmentation algorithm.*
>
> **A5**: Thank you very much for your interest and acknowledgment of our work. As stated in the abstract, we plan to open-source the training and testing code.
>
> We hope you are satisfied with our response and we welcome more feedback.

---

> > ### Comment · Reviewer_wCU2 · 2024-11-29
> >
> > I have increased my score to 6, while maintaining my original assessment.

---

### Official Review · Reviewer_pZEH · 2024-11-02

**Soundness:** 3
**Presentation:** 4
**Contribution:** 4
**Rating:** 8
**Confidence:** 3

**Summary:**

This paper presents an approach for neuron segmentation in electron microscopy images by a query-based segmentation framework. Traditional methods often rely on predicting affinities followed by watershed algorithms, which can introduce artifacts and inefficiencies. The proposed method predicts affinities using a lightweight branch to obtain coarse neuron structure information, which is then used to construct Affinity-Guided Queries (AGQs). These queries, along with learnable queries, interact with image features to directly predict the final segmentation results, eliminating the need for watershed algorithms. The method achieves better accuracy and a 2–3× speedup in inference time.

**Strengths:**

1. Novel and intersting idea that uses a query-based framework to eliminate the need for watershed algorithms used in traditional methods. The proposed method greatly improves efficiency.
2. Very well written manuscript. The paper clearly articulates the shortcomings of existing methods and motivates the need for the query based approach.
3. Comprehensive ablation studies show the contributions of different modules.

**Weaknesses:**

1. Despite the successful and interesting use of AGQs, I have difficulties to see why similar results couldn't be achieved by inputting image features, predicted affinities, and learnable queries directly into the neuron decoder (with modified architecture). While I understand that predicted affinities are used to create AGQ, I wonder if the LQ could "attend" to the predicted affinity map to construct the necessary queries themselves. It appears that the current neuron decoder is attempting to "rescue" the low-quality segmentation by AGQ in $M_{coarse}$.
2. The choice of $a$ in ConnectedComponent algorithm in eq 8 is not reported and discussed.
3. The computational efficiency is evaluated in terms of inference latency. It would be great if the manuscript could educate the readers about the model's memory consumption, which is important for deployment.

**Questions:**

1. There are, in fact, parallel/CUDA implementations of watershed algorithms. How would the proposed method compare to these optimized implementations in terms of efficiency?
2. For the Transformer Decoder implementation in Table 3, how do you input the image features and queries?

---

> ### Author Response · Authors · 2024-11-23
> **Response to reviewer pZEH**
>
> We thank you for your valuable reviews and address your concerns as follows.
>
> **Q1**: *Despite the successful and interesting use of AGQs, I have difficulties to see why similar results couldn't be achieved by inputting image features, predicted affinities, and learnable queries directly into the neuron decoder (with modified architecture). While I understand that predicted affinities are used to create AGQ, I wonder if the LQ could "attend" to the predicted affinity map to construct the necessary queries themselves. It appears that the current neuron decoder is attempting to "rescue" the low-quality segmentation by AGQ in M_coarse.*
>
> **A1**: We performed this experiment in Table 5 (*Concat affinities*) and found that feeding image features, predicted affinities, and learnable queries directly into the decoder achieved sub-optimal results. We believe the reason is that it might be difficult for the model to learn such an attention, e.g., the model could struggle to learn the connected components operation. Instead, our method integrates this geometric operation, which brings inductive bias to the model and reduces the learning difficulty of the decoder.
>
>
> **Q2**: *The choice of a in Connected Component algorithm in eq 8 is not reported and discussed.*
>
> **A2**: We clarify that a in Eq. 8 is the average value of affinities, rather than a fixed value. The motivation is to ensure that there is at least one foreground connected component (i.e., at least one affinity-guided query), thus avoiding training breakdown. Thank you for your suggestion, we have made this clearer in the paper.
>
>
> **Q3**: *The computational efficiency is evaluated in terms of inference latency. It would be great if the manuscript could educate the readers about the model's memory consumption, which is important for deployment.*
>
> **A3**: Thank you for your advice. We found that the model consumed about 6.5G of GPU memory and 5.4G of CPU memory for inference (with the input size of 17x257x257 and the volume of 100x1024x1024 used in the experiments), thus our model is deployable on most consumer GPU. We have included this discussion in the paper.
>
> **Q4**: *There are, in fact, parallel/CUDA implementations of watershed algorithms. How would the proposed method compare to these optimized implementations in terms of efficiency?*
>
> **A4**: Thank you for pointing this out. After investigating the relevant algorithms in detail, we found that there are no well-established CUDA implementations of the watershed algorithm available for comparison, and existing neuron segmentation frameworks also do not employ such algorithms [X1-X3]. The parallelizable watershed is still being explored by very recent efforts [X4]. On the other hand, even with the parallelizable watershed algorithm, it still faces problems such as producing excessive fragments, which would slow down the subsequent agglomeration step, as shown in Table 1. Conversely, the efficiency of our algorithm can also be further optimized for the implementation, e.g., processing multiple blocks in parallel, using compiled model (e.g., with tensorRT), etc. Basically, we regard our approach and the optimized implementations of watersheds as two different ways of addressing the efficiency problem within previous affinity-based methods.
>
> [X1] PyTorch Connectomics: A Scalable and Flexible Segmentation Framework for EM Connectomics. Arxiv preprint, 2021.
>
> [X2] Local shape descriptors for neuron segmentation. Nature Methods, 2023.
>
> [X3] SegNeuron: 3D Neuron Instance Segmentation in Any EM Volume with a Generalist Model. MICCAI, 2024.
>
> [X4] Parallel Watershed Partitioning: GPU-Based Hierarchical Image Segmentation. Arxiv preprint, 2024. (released at 14 Oct 2024)
>
> **Q5**: *For the Transformer Decoder implementation in Table 3, how do you input the image features and queries?*
>
> **A5**: For this implementation, within the cross-attention layer in the Transformer decoder, we treat the queries as Q, and the image features as K and V, following the practice in MaskFormer [X5].
>
> [X5] Per-pixel classification is not all you need for semantic segmentation. NeurIPS, 2021.
>
>
> We hope you are satisfied with our clarification above. Welcome any further comments, and we'd be happy to answer them in the discussion period.

---

> > ### Comment · Reviewer_pZEH · 2024-12-02
> >
> > Thanks for the detailed response, I maintain my original assessment.

---

### Official Review · Reviewer_nQ34 · 2024-11-03

**Soundness:** 2
**Presentation:** 3
**Contribution:** 2
**Rating:** 5
**Confidence:** 5

**Summary:**

This paper introduces Affinity-Guided Queries (AGQ) as a way to apply query-based instance segmentation to the field of connectomics. AGQ predicts the segments directly, bypassing postprocessing steps such as watershed used in some other approaches. The method is evaluated on two small-scale datasets, showing good results in terms of efficiency and accuracy.

**Strengths:**

- Evaluation on two datasets acquired with different EM techniques.
- Novel approach to segmentation of 3d EM data (query-based model).
- Good results in terms of efficiency and promising accuracy (but see comments about metrics below).
- Ablations showing the impact of the various modules.
- The paper is clearly written and well-organized.

**Weaknesses:**

- Evaluation is performed on a very small scale (tens of um^3). Experience in the field of connectomics shows that such results are often not predictive of reconstruction quality over larger volumes. One of the datasets (ZEBRAFINCH) provides full volume (1M um^3 -scale) skeleton tracings that make it possible to compute topological metrics, which can then be compared to those reported https://doi.org/10.1038/s41592-022-01711-z and https://doi.org/10.1038/s41592-018-0049-4. Having these available, would make it much clearer how the proposed method compares to the state of the art in a practical setting.
- VOI advantage seems to be driven by VOI_split reduction at the cost of VOI_merge. If the mergers happen at the level of supervoxels, the results are in practice less useful for proofreading, where many systems do not make it possible to manually fix this type of error.
- The model is quite architecturally complex, with different losses, modules, and an explicit affinity modeling branch.

**Questions:**

- Please include VOI_split and VOI_merge breakdown in the main text instead of only in the appendices.
- The intro, abstract and Fig. 1 make it sound as if all currently used neuron reconstruction methods rely on watershed. This is not true (the authors are aware of at least https://doi.org/10.1038/s41592-018-0049-4, which they cite later; there also exist other approaches such as https://openaccess.thecvf.com/content_CVPR_2019/html/Meirovitch_Cross-Classification_Clustering_An_Efficient_Multi-Object_Tracking_Technique_for_3-D_Instance_CVPR_2019_paper.html) and should be more clearly discussed earlier in the paper. The proposed model is far from from the first approach that directly converts 3d EM images into a segmentation.
- 0.08 mm^3 is a fairly specific number to quote as a "typical size of studied volume", and quotes a 6y old paper. It would be best to cite more works here, or just refrain from making statements about what is typical altogether. I would also suggest reformulating the comment about processing requiring "130 days of runtime", as this is highly specific to your specific cluster/hardware configuration.
- How often does your algorithm generate disconnected components for the same predicted object? (Eq. 3 does not guarantee connectivity)
- What's the motivation for predicting affinities and then immediately averaging them in Eq. 7? Why not predict a boundary map?
- Does N_a in Eq. 8 vary between passes through the network? If so, how does that interact the number of learned queries -- is N_l or N kept constant? How is N_l chosen in your work?
- Your method does not require block overlaps to generate consistent results. But wouldn't the additional spatial context help in making the predictions better? Have you explored this?
- What are the error bars for your results in Tab. 1?
- How does your network capacity (parameter count, FLOPS) compare with your baselines?
- Your examples in Fig. 5 focus on dendrites with broken off spines. Could you please also provide a representative figure for axons? (perhaps in the supplement)

---

> ### Author Response · Authors · 2024-11-23
> **Response to reviewer nQ34 (1/3)**
>
> We thank you for your valuable reviews and address your concerns as follows.
>
> **Q1**: *Evaluation is performed on a very small scale (tens of um^3). Experience in the field of connectomics shows that such results are often not predictive of reconstruction quality over larger volumes. One of the datasets (ZEBRAFINCH) provides full volume (1M um^3 -scale) skeleton tracings that make it possible to compute topological metrics, which can then be compared to those reported. Having these available, would make it much clearer how the proposed method compares to the state of the art in a practical setting.*
>
> **A1**: The mainly used test set in our paper (AC3) is about one hundred um^3, which is a commonly used public dataset in previous works [X3-X6]. We agree with you that validation on large-scale datasets can better exhibit the advantages of the proposed method and is more compatible with the needs of connectomics research, as the mentioned two Nature Methods papers [X1-X2] did. However, it is not trivial to process such a large scale of data. For example, hundreds of GPU hours are required for purely inference the model on the full volume of ZEBRAFINCH [X1]. Moreover, due to the spatial continuity of the 3D volume, it is not sufficient by simply processing at each local blocks, but requires some global post-processing to recover the morphology of the neurons. To complete such a project, it usually involves a huge amount of effort beyond reconstruction algorithm development, e.g., engineering techniques to efficiently save, process, evaluate the large volume data. For example, it is impractical to process the complete results (terabytes in size) in memory, requiring a complex distributed method to handle the volume. We clarify that our work mainly focuses on exploring new neuron segmentation algorithms. We believe the comparisons for smaller datasets are also meaningful and would help us design better algorithms, which has also been widely adopted in previous work [X3-X7]. We appreciate your insightful suggestion and will take it as the next step of our future research.
>
> [X1] Local shape descriptors for neuron segmentation. Nature Methods, 2023.
>
> [X2] High-precision automated reconstruction of neurons with flood-filling networks. Nature Methods, 2018.
>
> [X3] Learning to Model Pixel-Embedded Affinity for Homogeneous Instance Segmentation. AAAI, 2023.
>
> [X4] Electron microscopy images as set of fragments for mitochondrial segmentation. AAAI, 2024.
>
> [X5] Self-Supervised Neuron Segmentation with Multi-Agent Reinforcement Learning. IJCAI, 2023.
>
> [X6] Cross-dimension affinity distillation for 3d EM neuron segmentation. CVPR, 2024.
>
> [X7] Large Scale Image Segmentation with Structured Loss Based Deep Learning for Connectome Reconstruction. TPAMI, 2019.
>
>
> **Q2**: *VOI advantage seems to be driven by VOI_split reduction at the cost of VOI_merge. If the mergers happen at the level of supervoxels, the results are in practice less useful for proofreading, where many systems do not make it possible to manually fix this type of error.*
>
> **A2**: Thank you for pointing out this practical problem. We respond to your concern in the following two aspects:
>
> -	First, a considerable number of mergers occur at the agglomeration step rather than at the supervoxel-level. For example, the VOI_merge before the agglomeration is only 0.181 on the AC3 dataset, which is smaller than all the baselines. This part of mergers (beyond supervoxel level) can be proofread by, for example, adjusting the threshold for agglomeration, similar to existing proofreading systems.
> -	Second, existing proofreading systems are highly coupled with the traditional affinity-based methods, so it would be unfair to criticize our method based on these systems. As a new neuron segmentation paradigm, our method might require new proofreading algorithms, which we regard as a promising future work. For example, a possible solution is local re-segmentation during proofreading, e.g., run watershed locally (note that our model also predicts affinity) or use interactive segmentation models (e.g., SAM [X8]) to eliminate merge errors at the supervoxel level. Since these mergers are limited to small blocks (e.g., 17x257x257) for our method, the local re-segmentation would not introduce too much overhead.
>
> [X8] Segment Anything. ICCV, 2023.

---

> ### Author Response · Authors · 2024-11-23
> **Response to reviewer nQ34 (2/3)**
>
> **Q3**: *The model is quite architecturally complex, with different losses, modules, and an explicit affinity modeling branch.*
>
> **A3**: Besides the foundational 3D U-Net backbone, our model integrates a neuron decoder and a query generator, succinctly illustrated in Figure 2(c). The segmentation and affinity losses are essential to supervise the two modules respectively, while the contrastive loss is to aid in the learning of the subsequent modules. It is important to note that previous affinity-based methods require more complex steps to obtain the final results, compared to which our method is significantly more concise and efficient during inference.
>
>
> **Q4**: *Please include VOI_split and VOI_merge breakdown in the main text instead of only in the appendices.*
>
> **A4**: Thanks to your suggestion, we have reorganized Table 1 to include VOI_split and VOI_merge breakdown.
>
>
> **Q5**: *The intro, abstract and Fig. 1 make it sound as if all currently used neuron reconstruction methods rely on watershed. This is not true (the authors are aware of at least FFN, which they cite later; there also exist other approaches such as 3C) and should be more clearly discussed earlier in the paper. The proposed model is far from the first approach that directly converts 3d EM images into a segmentation.*
>
> **A5**: Thank you for pointing out this issue and this related work (subsequently denoted as 3C). To clarify, our paper does not state that all previous methods rely on watershed, nor that we are the first to directly convert 3D EM images to segmentation. We consider 3C to be more akin to a learnable agglomeration algorithm, as it requires the fragments to be provided as seeds and can only merges these fragments for a more complete result. Instead, our method focuses on developing a query-based paradigm that bridges neuron segmentation with modern natural image segmentation and allows it to benefit from the advances in natural images segmentation methods. As per your suggestion, we have discussed these efforts earlier in the paper.
>
>
> **Q6**: *0.08 mm^3 is a fairly specific number to quote as a "typical size of studied volume", and quotes a 6y old paper. It would be best to cite more works here, or just refrain from making statements about what is typical altogether. I would also suggest reformulating the comment about processing requiring "130 days of runtime", as this is highly specific to your specific cluster/hardware configuration.*
>
> **A6**: Thank you for your suggestion, we have cited more recent works [X9-X10] and revised the statements.
>
> [X9] The connectome of an insect brain. Science, 2023.
>
> [X10] A petavoxel fragment of human cerebral cortex reconstructed at nanoscale resolution. Science, 2024.
>
> **Q7**: *How often does your algorithm generate disconnected components for the same predicted object? (Eq. 3 does not guarantee connectivity)*
>
> **A7**: You are correct that our method does predict disconnected components in some cases (about 0.7% voxels). These disconnected components usually appear as small fragments and can be easily filtered out by, for example, a connected component algorithm and a volume threshold. A failure case is shown in region A of Figure 10 and discussed in Appendix G.
>
>
> **Q8**: *What's the motivation for predicting affinities and then immediately averaging them in Eq. 7? Why not predict a boundary map?*
>
> **A8**: Our motivation is mainly based on the following two considerations:
>
> -	The affinity is more well-defined between objects. Notably, existing neuron segmentation annotations allow the masks of different neurons to be directly touched, with no background voxels serving as boundaries. Leveraging affinities aligns better with this annotation style.
>
> -	Furthermore, predicting affinity is more compatible with existing agglomeration algorithms. As elaborated in Section C.1, utilizing affinities simplifies the integration of established agglomeration functions for merging results across various blocks.
>
>
> **Q9**: *Does N_a in Eq. 8 vary between passes through the network? If so, how does that interact the number of learned queries -- is N_l or N kept constant? How is N_l chosen in your work?*
>
> **A9**: For the queries, $N_a$ (number of affinity-guided queries) vary between passes through the network while $N_l$ (number of learned queries)  kept constant. $N$ (= $N_a$ + $N_l$) varies with $N_a$. We have made it clearer in the paper.

---

> ### Author Response · Authors · 2024-11-23
> **Response to reviewer nQ34 (3/3)**
>
> **Q10**: *Your method does not require block overlaps to generate consistent results. But wouldn't the additional spatial context help in making the predictions better? Have you explored this?*
>
> **A10**: We have explored this preliminarily. However, for our model, it is not trivial to use overlap predictions to improve the results. Because the output of our model is at the instance level and instance ids may not be consistent across blocks, it is not straightforward to “average” the results of the overlapped regions between blocks. We have tried an FFN-style approach, i.e., accepting all splits, but it makes the pipeline more complicated and the improvement is marginal. Consequently, we have opted against incorporating block overlaps into our method.
>
>
> **Q11**: *What are the error bars for your results in Tab. 1?*
>
> **A11**: Thank you for pointing out this issue. We have re-executed the results in Table 1 multiple times and calculated the error bar, which is supplemented in the table below. Note that some of models in Table 1 are not included as there is no available code (see Table 1 for details).
>
> |          |    VOI_split |  VOI_merge |  VOI  |   Arand  |
> |:----------:|:-----------:|:----------:|:--------:|:----------:|
> | ResUNet  | 1.026 $\pm$ 0.015 | 0.257 $\pm$ 0.001 | 1.284 $\pm$ 0.016| 0.151 $\pm$ 0.004|
> | SeUNet   | 1.030 $\pm$ 0.009 | 0.251 $\pm$ 0.001 | 1.280 $\pm$ 0.009| 0.151 $\pm$ 0.006|
> | SwinUNETR| 1.219 $\pm$ 0.027 | 0.250 $\pm$ 0.084 | 1.469 $\pm$ 0.056| 0.135 $\pm$ 0.036|
> | UNETR    | 2.753 $\pm$ 0.004 | 0.314 $\pm$ 0.046 | 3.067 $\pm$ 0.051| 0.233 $\pm$ 0.018|
> | SuperHuman|1.116 $\pm$ 0.042 | 0.313 $\pm$ 0.071 | 1.429 $\pm$ 0.029| 0.141 $\pm$ 0.027|
> | LSD      | 1.407 $\pm$ 0.058 | 0.311 $\pm$ 0.116 | 1.717 $\pm$ 0.057| 0.143 $\pm$ 0.013|
> | PEA      | 0.871 $\pm$ 0.028 | 0.261 $\pm$ 0.042 | 1.133 $\pm$ 0.069| **0.098** $\pm$ 0.006|
> | AGQ (ours)|0.688 $\pm$ 0.015 | 0.285 $\pm$ 0.016 | **0.972** $\pm$ 0.029| **0.098** $\pm$ 0.011|
>
>
> **Q12**: *How does your network capacity (parameter count, FLOPS) compare with your baselines?*
>
> **A12**: We have added the following table, where our method introduced a certain amount of computation and a small number of parameters than the 3D ResUNet backbone. Note that this paper focuses on latency and does not perform tailored optimization on FLOPs. We found that the FLOPs highly concentrated in the affinity branch (about 126.7 GFLOPs), and targeted architecture optimization could further reduce the FLOPs.
>
> |          |  VOI  | Arand | Infer. Time / s | FLOPS / G | #Params / M |
> |----------|-------|-------|------------------|-----------|-------------|
> | ResUNet  | 1.295 | 0.154 | 147.8            | 141.9    | 3.5         |
> | SeUNet   | 1.282 | 0.156 | 149.8            | 142.0    | 3.5         |
> | UNETR    | 3.031 | 0.220 | 211.4            | 199.8    | 90.7        |
> | SwinUNETR| 1.429 | 0.110 | 155.6            | 724.0   | 62.2        |
> | MALA     | 1.546 | 0.120 | -                | -         | 84.0        |
> | FragViT  | 1.054 | 0.093 | >122.7           | -         | 34.2        |
> | AGQ (ours)| 0.967 | 0.095 | 33.7             | 412.2     | 4.7         |
>
>
> **Q13**. *Your examples in Fig. 5 focus on dendrites with broken off spines. Could you please also provide a representative figure for axons? (perhaps in the supplement)*
>
> **A13**: Thanks to your suggestion, we have added this visualization in Figure 12 and discussed the results in Appendix J.
>
> We hope you are satisfied with our clarification above. Welcome any further comments, and we'd be happy to answer them in the discussion period.

---

> > ### Comment · Reviewer_nQ34 · 2024-11-27
> >
> > Thank you for the detailed response.
> >
> > I appreciate the increased complexity and computational cost of running evaluations over larger volumes. However, in my experience in the field, evaluations over small subvolumes like the one used in this work simply do not generalize to larger volumes. For any proposed method to be practically relevant, it is critical that the evaluation at a relevant scale. While a full run on ZEBRAFINCH might not be needed, you could consider evaluating on subvolumes of ZEBRAFINCH, FIB25 or HEMIBRAIN similarly to prior work (e.g. the LSD paper).
> >
> > I also don't agree that existing proofreading systems are necessarily limited by or designed for affinity methods. Various heuristic or learned techniques exist to recompute the shapes locally, but are in practice rarely used because of the infrastructural complexities involved in updating the segmentation volumes.
> >
> > Thank you for adding appendix J. However, I was hoping you could include examples of thin, non-myelinated axons, illustrating impact on continuity, not local morphology details. The fact that myelinated axons appear hard to segment is likely a result of their insufficient representation in the training data. Given their relatively large caliber, they are unlikely to present any intrinsic segmentation difficulties.
> >
> > Additional minor requests:
> > - Please include VOI_m and VOI_s for both pre- and post-agglomeration states?
> > - Please include the VOI_m/VOI_s breakdown in all tables (3, 5, 7, 8, 9)? These two metrics are in practice much more informative than Arand, so I would suggest dropping the latter in favor of the two VOI scores where space is an issue.
> > - Regarding A11: what was varied between runs to compute these CIs?
> > - How is $N_a$ determined?

---

> ### Author Response · Authors · 2024-11-30
> **Response to reviewer nQ34**
>
> Thank you for your valuable feedback and acknowledgment of our efforts. Your comments are greatly appreciated and encourage us to continue refining our work.
>
> **Q1**: *I appreciate the increased complexity and computational cost of running evaluations over larger volumes. However, in my experience in the field, evaluations over small subvolumes like the one used in this work simply do not generalize to larger volumes. For any proposed method to be practically relevant, it is critical that the evaluation at a relevant scale. While a full run on ZEBRAFINCH might not be needed, you could consider evaluating on subvolumes of ZEBRAFINCH, FIB25 or HEMIBRAIN similarly to prior work (e.g. the LSD paper).*
>
> **A1**: We would like to clarify that the focus of this work is on exploring the feasibility of a query-based framework for neuron segmentation and revealing its potential for efficiency. To this end, we followed the experimental setup used in previous papers [X3-X7] within the computer vision community. Specifically, we used the AC3/AC4 dataset and subvolumes of ZEBRAFINCH for our evaluation. Evaluating larger-scale volumes presents engineering challenges, as well as the need for specialized block assembly algorithms to better adapt to these larger volumes. We appreciate your suggestions and will certainly prioritize larger-scale evaluations in future work.
>
> **Q2**: *I also don't agree that existing proofreading systems are necessarily limited by or designed for affinity methods. Various heuristic or learned techniques exist to recompute the shapes locally, but are in practice rarely used because of the infrastructural complexities involved in updating the segmentation volumes.*
>
> **A2**: We would like to clarify that our emphasis is on the possibility for manually correcting these errors. While we acknowledge that some heuristic or learned techniques are already available, we also hold the view that as proofreading systems evolve, the infrastructural challenges involved can be overcome. Furthermore, we believe that the development of  query-based approaches, such as ours, will better drive this progress, as it introduces new demands for proofreading algorithms.
>
> **Q3**: *Thank you for adding appendix J. However, I was hoping you could include examples of thin, non-myelinated axons, illustrating impact on continuity, not local morphology details. The fact that myelinated axons appear hard to segment is likely a result of their insufficient representation in the training data. Given their relatively large caliber, they are unlikely to present any intrinsic segmentation difficulties.*
>
> **A3**: We would like to clarify that our algorithm, as well as the baseline methods, does not differentiate between different types of neuronal structures. As a result, segmenting thin axons is not inherently more challenging than segmenting thin spines, as they share locally similar morphology. Therefore, we believe that the comparison shown in Figure 5 provides a meaningful demonstration of the model's ability to segment slender neuronal structures. We appreciate your suggestion and will add a comparison of different neuronal structural types, including various axons and dendrites, in the final version of the paper, since the revised paper cannot be submitted at this moment.
>
> **Q4**: *Please include VOI_m and VOI_s for both pre- and post-agglomeration states?*
>
> **A4**: Thank you for your suggestion. We have included the VOI_m and VOI_s results for both pre- and post-agglomeration states in the revised version (Table 6 in Appendix C.1).
>
> **Q5**: *Please include the VOI_m/VOI_s breakdown in all tables (3, 5, 7, 8, 9)? These two metrics are in practice much more informative than Arand, so I would suggest dropping the latter in favor of the two VOI scores where space is an issue.*
>
> **A5**: In response to your suggestion, we have updated Tables 3, 5, 7, 8, and 9 to include the VOI_m/VOI_s breakdown in the same format as Table 4.
>
> **Q6**: *Regarding A11: what was varied between runs to compute these CIs?*
>
> **A6**: We kept all hyperparameters fixed and only re-trained these models for evaluation. The variability in the results arises from factors such as different random seeds, numerical errors during training, and the non-deterministic algorithms of PyTorch.
>
> **Q7**: *How is Na determined?*
>
> **A7**: Na represents the number of connected components in the thresholded mean affinities (L298-301). To prevent GPU memory overflow during training, Na is capped at an upper limit of 100, which is greater than the number of connected components in most cases.
>
> We hope that our response addresses your concerns and are open to any further feedback you may have.

---

> > ### Comment · Reviewer_nQ34 · 2024-12-02
> >
> > Thank you for the detailed responses!
> >
> > I acknowledge the increased technical complexity involved in evaluating over larger datasets, but I think you overestimate the amount of effort that is really needed for this. Block assembly could be as simple as computing global connected components based on small overlaps, which can be done in memory on a single machine for the volume sizes we are considering here (e.g. a subset of ZEBRAFINCH).
> >
> > I think your paper is indeed an interesting exploration of whether query-based methods work for EM segmentation, and shows promising results. However, you also claim that your method is faster and improves over SOTA. In my experience, the latter is just not sufficiently supported given the spatial scales involved.
> >
> > Thanks again for the discussions here. After reviewing all the comments again, I decided to keep my original score.

---

### Official Review · Reviewer_uA9h · 2024-11-04

**Soundness:** 3
**Presentation:** 3
**Contribution:** 3
**Rating:** 6
**Confidence:** 4

**Summary:**

The paper presents a novel method for neuron segmentation in electron microscopy (EM) images using affinity-guided queries within a query-based framework. The approach aims to overcome the limitations of traditional methods that rely on watershed algorithms by directly predicting segmentation results, achieving improved accuracy and efficiency. The method is evaluated on benchmark datasets, showing competitive performance.

**Strengths:**

1. The use of affinity-guided queries in a query-based framework is a novel contribution to the field of neuron segmentation.
2. The method demonstrates a significant speedup (2-3x) over traditional methods, a notable advantage for large-scale data processing.
3. The query-based framework directly predicts the final segmentation results, avoiding the inaccuracy and inefficiency associated with the watershed algorithm used in traditional methods.

**Weaknesses:**

1. The comparison with state-of-the-art methods is limited to a few selected approaches. A more extensive comparison with a broader range of recent methods would strengthen the paper's contributions, such as [1].
2. The paper does not provide a strong theoretical foundation for why affinity-guided queries outperform other query-based methods, especially in terms of handling neuron morphology.  It is better to give a more detailed explanation or analysis of the mechanisms by which affinity-guided queries better capture neuron morphology compared to other query-based approaches.

[1] Zhang, Yanchao, et al. "SegNeuron: 3D Neuron Instance Segmentation in Any EM Volume with a Generalist Model." International Conference on Medical Image Computing and Computer-Assisted Intervention. Cham: Springer Nature Switzerland, 2024.

**Questions:**

1. How does the method handle variations in image quality or noise levels in EM datasets?
2. What are the specific computational requirements for training and inference, and how do these scale with larger volumes?
3. How do affinity-guided queries compare with other forms of guided queries in terms of performance and efficiency?
4. The paper mentions that the proposed method does not require overlap prediction to address border effects. However, it is unclear how the method handles the continuity and consistency of segmentation results across block boundaries. Can the authors elaborate on this aspect and provide visual examples?

---

> ### Author Response · Authors · 2024-11-23
> **Response to reviewer uA9h (1/2)**
>
> We thank you for your valuable reviews and address your concerns as follows.
>
> **Q1**: *The comparison with state-of-the-art methods is limited to a few selected approaches. A more extensive comparison with a broader range of recent methods would strengthen the paper's contributions, such as SegNeuron.*
>
> **A1**: Thanks for your suggestion, SegNeuron is an impressive work dedicated to generalized neuron segmentation, which is contemporary with our work. We have supplemented the comparison with SegNeuron on the AC3 dataset in the table below. **Note that one of the SegNeuron's training sets (*i.e.,* Kasthuri) contains AC3 (*i.e.,* our test set).** We found that even with this unfair comparison, SegNeuron achieved only slightly better VOI than ours. However, our method is 9 times faster than SegNeuron, demonstrating the strong efficiency advantage of our method.
>
>
> |          | VOI_split | VOI_merge |  VOI  | Arand | model (s) | watershed (s) |  agglormetation (s) | total (s) |
> |----------|-----------|-----------|-------|-------|-------|-----------|-------|-------|
> | SegNeuron|    0.698  |   0.245   | **0.943** | **0.088** | 249.7 |    27.5   |  30.1 | 307.3 |
> | AGQ (ours)|   0.677  |   0.290   | 0.967 | 0.095 |  27.6 |    N/A    |   6.1 |  **33.7** |
>
> **Q2**: *The paper does not provide a strong theoretical foundation for why affinity-guided queries outperform other query-based methods, especially in terms of handling neuron morphology. It is better to give a more detailed explanation or analysis of the mechanisms by which affinity-guided queries better capture neuron morphology compared to other query-based approaches.*
>
> **A2**: This is based on the two facts:
>
> - Our affinity-guided queries are constructed by multiplying and averaging the coarse masks with the image features (Section 3.5);
> - We applied a contrastive loss to the image features (Sections 3.6 and B.2).
>
> These ensure that our affinity-guided queries naturally bear a stronger similarity to the features of the corresponding neurons, and thus in more easily captures the morphological structure of the neurons. For example, just multiplying these queries on the feature map would result in an approximate segmentation ($P_0$ in Figure 6).
>
> **Q3**: *How does the method handle variations in image quality or noise levels in EM datasets?*
>
> **A3**: We applied various data augmentations during training to enhance the robustness of the model against image noise, following pytorch_connectomics code base:
>
> -	ELASTIC: applying random elastic deformation of images;
> -	MISSINGPARTS: randomly mask out some input regions;
> -	MISSINGSECTION: randomly mask out some input sections;
> -	MISALIGNMENT: randomly displace the pixels in a section;
> -	MOTIONBLUR: randomly apply motion blur;
> -	CUTBLUR: randomly blur some regions;
> -	CUTNOISE: randomly add noise to some regions;
> -	GRAYSCALE: randomly adjust the contrast/brightness and invert the color space and apply gamma correction;
> -	RESCALE: randomly scale the volume;
> -	ROTATE, FLIP: randomly rotate and flip the entire volume;
>
> These data augmentations cover most of the potential imaging variations and therefore allow our model to handle these noises during inference. Besides, our method can also be combined with denoising techniques [X1-X2] to further enhance its robustness against noises.
>
>
> [X1] Unsupervised Domain Adaptation for EM Image Denoising with Invertible Networks. TMI, 2024.
>
> [X2] Joint EM Image Denoising and Segmentation with Instance-aware Interaction. MICCAI, 2024.
>
> **Q4**: *What are the specific computational requirements for training and inference, and how do these scale with larger volumes?*
>
> **A4**: The training required about 40 hours on 8x 3090 GPUs while the inference took about 34 seconds on 1x 3090 GPU for a 100×1024×1024 volume. Notably, the inference time is proportional to the number of voxels as it is an O(n) time complexity algorithm. Presently, our training configuration uses a fixed number of iterations, making the training cost independent of the volume size. However, in scenarios where the training volume significantly increases, extending the training schedule could optimize the utilization of training data more effectively.

---

> ### Author Response · Authors · 2024-11-23
> **Response to reviewer uA9h (2/2)**
>
> **Q5**: *How do affinity-guided queries compare with other forms of guided queries in terms of performance and efficiency?*
>
> **A5**: In Section J.1, we conducted a comparative analysis of various methods for generating guided queries, including watershed and connected components. Our investigation revealed that the proposed affinity-guided queries achieved better results. While exploring the use of clustering techniques, such as k-means, to derive guided queries, we encountered an unacceptable computational delay. For example, in our model, the k-means required 240 seconds to produce 100 queries, whereas the affinity-guided approach (specifically, connected components) took less than 40 milliseconds. These suggest that the affinity-guided queries used is better than its counterparts, both in terms of effectiveness and speed.
>
> **Q6**: *The paper mentions that the proposed method does not require overlap prediction to address border effects. However, it is unclear how the method handles the continuity and consistency of segmentation results across block boundaries. Can the authors elaborate on this aspect and provide visual examples?*
>
> **A6**: It is essential to note that in affinity-based methods, misaligned affinities can significantly impact the performance of the watershed algorithm, leading to terrible results (refer to Figure 9). However, this issue has a relatively minor effect on our algorithm, which does not rely on watersheds. For our algorithm, misaligned segmentation may lead to agglomeration failures, potentially resulting in split errors (e.g., as seen in region B in Figure 10). Additionally, as depicted in Figures 5 and 8, our method consistently demonstrates minimal inconsistencies along block boundaries in most cases.
>
>
> We hope you are satisfied with our clarification above. Welcome any further comments, and we'd be happy to answer them in the discussion period.

---

> > ### Comment · Reviewer_uA9h · 2024-12-01
> >
> > Thank you for the rebuttal. After carefully reviewing your responses, I have decided to maintain my score.

---

### Author Response · Authors · 2024-11-23
**General Response**

The authors sincerely appreciate the time and effort the reviewers put into this manuscript, which helps us a lot to improve the quality of the revised paper. It is pretty encouraging that the reviewers found the proposed method is *novel* [uA9h, nQ34, pZEH] and *interesting* [pZEH], and is *commendable and addresses significant challenges in the field* [wCU2], the experiments showed *a notable advantage* [uA9h] and *good results in terms of efficiency and promising accuracy* [nQ34], with *comprehensive ablation studies* [pZEH], and the written is *clearly* and *well-organized* [nQ34]. We carefully address the concerns below, and the paper has been revised accordingly (highlighted in blue).

---

### Meta-Review · Area_Chair_TSmx · 2024-12-21

**Metareview:**

This paper proposes a network architecture for instance segmentation of electron microscopy (EM) images. Unlike previous methods with two stages of affinity prediction and watershed-based segmentation, the method here is end-to-end with several losses minimized. Unlike standard query-based segmentation, the network here includes a new module of affinity-based queries. The proposed pipeline is technically sound and is efficient during inference compared to standard query-based segmentation networks.
Three reviewers gave positive ratings with one reviewer giving a negative rating. For the review with a negative rating, the major concerns are about the complexity of the pipeline with several losses and the lack of experiments on large-scale 3D volume. However, other reviewers didn't find the proposed pipeline too complex or impractical. Also, the results on medium-scale 3D datasets demonstrated the effectiveness and efficiency of the proposed method.
Overall, the strengths overweight the weaknesses by a large margin, and the AC recommends acceptance of this paper.

**Additional Comments On Reviewer Discussion:**

Reviewer uA9h and reviewer wCU2 had concerns about the lack of comparison to recent work. The authors have provided additional experiment results comparing to the suggested paper.
Review uA9h  and reviewer pZEH questioned the design of affinity-guided queries and even suggested alternative design choices. The authors have further clarified the motivation and advantages of affinity-guided queries and included additional results of other variant architectures.

---

### Decision · Program_Chairs · 2025-01-22

Accept (Poster)